# Inference of RNA decay rate from transcriptional profiling highlights the regulatory programs of Alzheimer's disease

Rached Alkallas[1,2], Lisa Fish[3,4,5], Hani Goodarzi[3,4,5] & Hamed S. Najafabadi [1,2]

The abundance of mRNA is mainly determined by the rates of RNA transcription and decay. Here, we present a method for unbiased estimation of differential mRNA decay rate from RNA-sequencing data by modeling the kinetics of mRNA metabolism. We show that in all primary human tissues tested, and particularly in the central nervous system, many pathways are regulated at the mRNA stability level. We present a parsimonious regulatory model consisting of two RNA-binding proteins and four microRNAs that modulate the mRNA stability landscape of the brain, which suggests a new link between RBFOX proteins and Alzheimer's disease. We show that downregulation of RBFOX1 leads to destabilization of mRNAs encoding for synaptic transmission proteins, which may contribute to the loss of synaptic function in Alzheimer's disease. RBFOX1 downregulation is more likely to occur in older and female individuals, consistent with the association of Alzheimer's disease with age and gender.

---

[1] Department of Human Genetics, McGill University, Montreal, QC, Canada H3A 0C7. [2] McGill University and Genome Quebec Innovation Centre, Montreal, QC, Canada H3A 0G1. [3] Department of Biochemistry and Biophysics, University of California, San Francisco, CA 94158, USA. [4] Department of Urology, University of California, San Francisco, CA 94158, USA. [5] Helen Diller Family Comprehensive Cancer Center, University of California, San Francisco, CA 94158, USA. Rached Alkallas and Hamed S. Najafabadi contributed equally to this work. Correspondence and requests for materials should be addressed to H.S.N. (email: hamed.najafabadi@mcgill.ca)

The mRNA decay rate is a key determinant of steady-state mRNA abundance. Measuring the mRNA decay rate generally involves time series examination of mRNA abundance following blockage of transcription[1,2], or monitoring the abundance of labeled mRNA using high-throughput pulse-chase methods[3,4]. These methods are mostly suitable for analysis of cell cultures, however, and cannot be used for measuring mRNA decay rate in tissue samples.

In principle, it should be possible to measure the decay rate of mRNAs by decoupling the transcription rate from the steady-state mRNA abundance. It has been suggested that, while exonic read counts in RNA-seq data correspond to steady-state mRNA abundance, changes in the abundance of intronic reads can be used to estimate the change in transcription rate[5–7]. Therefore, RNA-seq reflects a snapshot of both steady-state mRNA level and transcriptional activity, providing the possibility for deconvolving the mRNA decay rate from RNA-seq data.

This concept has been recently suggested to capture the differential mRNA decay rate in a wide set of contexts[5], where the change in mRNA half-life is estimated as the difference of the logarithm of fold-change of exonic reads and the logarithm of fold-change of intronic reads ($\Delta$exon–$\Delta$intron). Here, we show that this measure is highly biased, and generally overestimates the stability of slow-transcribing mRNAs and underestimates the stability of fast-transcribing mRNAs. We propose a generalizable approach for correcting this bias, and use it to investigate the post-transcriptional regulatory programs of a panel of human tissues.

Our results indicate that mRNA stability plays an integral role in shaping the transcriptomes of all tested tissues, particularly those of the central nervous system. We show that a substantial portion of the brain mRNA stability profile can be explained by the functions of two RNA-binding protein families (the RBFOX and ZFP36 families) and four miRNAs (miR-124, miR-29, miR-9, and miR-128). The RBFOX targets are enriched for mRNAs that encode synaptic transmission proteins, and are destabilized in the brains of individuals with Alzheimer's disease (AD). We show that knockdown of RBFOX1 can partially recreate the AD stability profile, and its expression rescues the normal transcriptome, suggesting a link between dysregulation of RBFOX1 and AD.

## Results

**Decoupling of changes in transcription and mRNA decay rates.** Several recent studies have used the difference of the logarithm of fold-change of exonic reads and the logarithm of fold-change of intronic reads ($\Delta$exon–$\Delta$intron) as an estimate of the change in mRNA half-life[5,8]. However, by considering a simple model of RNA metabolism (Fig. 1a) and solving the kinetic equations of this model (Methods and Supplementary Fig. 1), we found that $\Delta$exon–$\Delta$intron is affected by a "bias term" that depends on the maximum capacity of the pre-mRNA processing machinery, which is limited in the cell[9], as well as the change in transcription rate (Fig. 1b). The presence of this bias leads to a negative correlation between $\Delta$exon–$\Delta$intron and differential transcription rate. In other words, even in the absence of any change in mRNA stability, $\Delta$exon–$\Delta$intron would still be positive for genes that were transcriptionally downregulated, and negative for genes that were transcriptionally upregulated (Fig. 1c).

To test this model, we measured $\Delta$exon–$\Delta$intron for human genes using RNA-seq data from a panel of 20 human tissues[10]. Indeed, we observed a significant negative trend between $\Delta$exon–$\Delta$intron and the change in transcription rate (Fig. 1d and Supplementary Fig. 2a), in agreement with our model. Other data sets that we analyzed also showed the same trend, including RNA-seq data from Illumina BodyMap 2.0, data from nine

human cell lines from ENCODE[11], and three RNA-seq datasets from mouse[12–14] (Supplementary Fig. 2a, b), suggesting that this bias is not data-set-specific or species-specific. This correlation was not due to inaccuracies in estimating $\Delta$exon and $\Delta$intron from low read counts, since a negative correlation could be observed for gene sets with varying read coverage (Fig. 1e). In fact, higher expression was associated with larger magnitudes of bias (Fig. 1e), consistent with our kinetic model in which saturation of the pre-mRNA processing machinery results in a steeper bias (Fig. 1c). This bias in $\Delta$exon–$\Delta$intron can substantially confound downstream analysis of mRNA stability programs. For example, in several tissues, pathways that are transcriptionally downregulated generally appear to be stabilized at the mRNA level (Supplementary Fig. 2c), and in other tissues the bias in $\Delta$exon–$\Delta$intron partially masks the synergy between transcriptional and post-transcriptional regulation of pathways (Supplementary Fig. 2d).

We devised a computational framework that estimates the transcription rate-dependent bias from RNA-seq data and subtracts it from $\Delta$exon–$\Delta$intron (Methods, Fig. 2a and Supplementary Fig. 3), providing unbiased estimates of differential mRNA half-life. Although this framework makes simplifying assumptions, such as invariability of some of the kinetic parameters of mRNA metabolism (see Methods), our simulations suggest that even when these assumptions are not met, this framework should still provide more accurate estimates of mRNA stability compared to uncorrected $\Delta$exon–$\Delta$intron (Supplementary Fig. 4). To empirically evaluate our method, we analyzed the RNA-seq data[15] from the breast carcinoma cell line MDA and the in vivo-selected highly metastatic sub-line MDA-LM2[16]. We analyzed this dataset because a large number of mRNAs were previously identified as stabilized or destabilized in MDA-LM2 cells compared to the parental MDA line[17] using BRIC-seq[18]. Our analysis showed that for highly biased genes, uncorrected $\Delta$exon–$\Delta$intron is not an accurate estimate of differential stability, as expected from our kinetic model. In fact, when the magnitude of the bias term was large, uncorrected $\Delta$exon–$\Delta$intron was on average higher for LM2-destabilized genes and lower for LM2-stabilized genes (Fig. 2b). In contrast, our framework provided estimates that were overall consistent with the BRIC-seq differential stability measurements, even for highly biased genes (Fig. 2b and Supplementary Fig. 5a, b). To further evaluate our method, we selected six additional genes that did not have statistically significant differences in BRIC-seq measurements between MDA-parental and MDA-LM2 cells, and had the largest bias based on analysis of RNA-seq data (see Methods): three of these genes appeared to be significantly more stable in LM2 cells based on $\Delta$exon–$\Delta$intron, but after bias correction were inferred to be more stable in MDA cells; the other three genes showed the reverse pattern. We measured the stability of these genes in MDA-parental and MDA-LM2 cells by α-amanitin inhibition of transcription followed by qRT-PCR. Of the six genes that we examined, four genes indeed showed significant differential stability ($P < 0.05$). In all four cases, the direction of the change in stability was consistent with the bias-corrected predictions but not with the uncorrected $\Delta$exon–$\Delta$intron (Fig. 2c). Additional analysis for comparison of $\Delta$exon–$\Delta$intron and our unbiased estimates of stability can be found in Supplementary Fig. 5c, d. We have implemented our computational framework in a software package available at https://github.com/csglab/REMBRANDTS.

**The mRNA stability programs of human brain.** We sought to explore the post-transcriptional programs across human tissues using our unbiased estimate of mRNA stability. We analyzed the

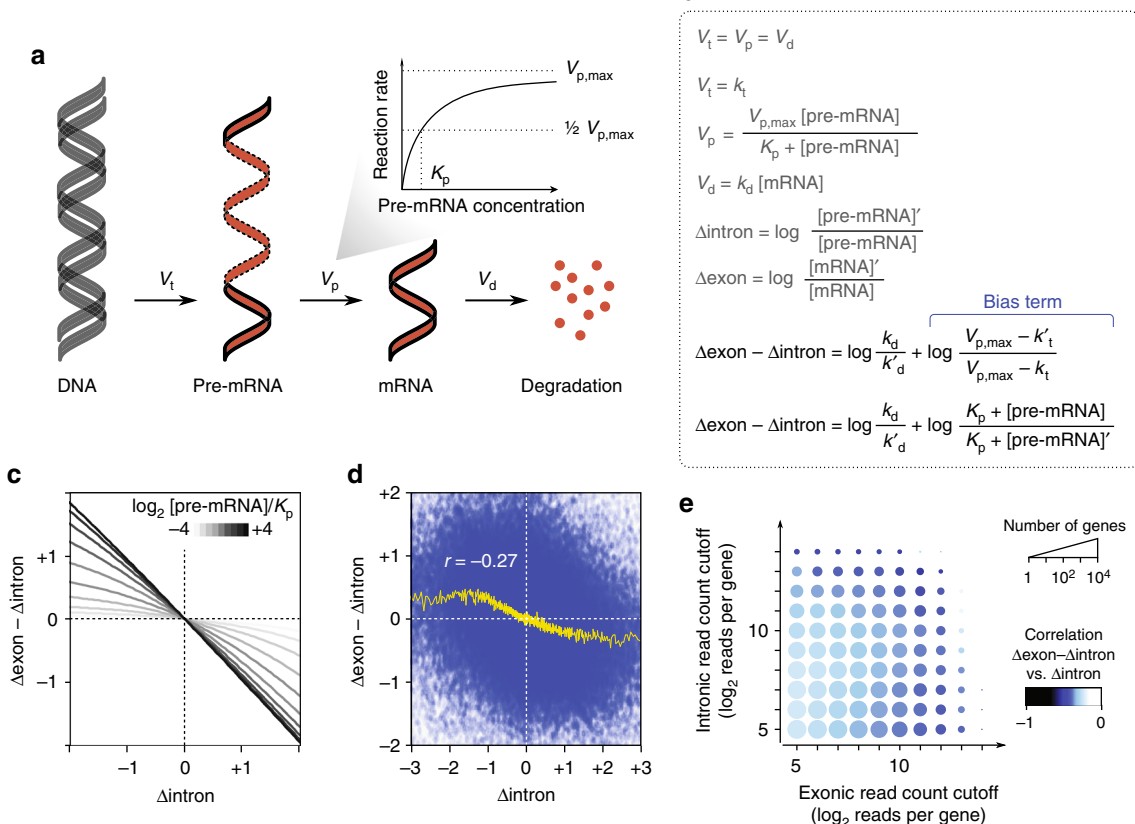

**Fig. 1** Δexon–Δintron is a biased estimate of mRNA stability. **a** A simplified schematic model of mRNA transcription and processing. The rate of RNA processing is modeled based on Michaelis–Menten kinetics, where the maximum rate is largely determined by the availability of the splicing machinery. $V_t$: transcription rate, $V_p$: RNA processing rate, $V_d$: RNA degradation rate, $K_p$: Michaelis constant for RNA processing. **b** The continuity and rate equations for RNA processing reactions, and the resulting equation for the relationship between Δexon–Δintron and the rates of different steps. $k_t$: transcription rate constant, $k_d$: mRNA decay rate constant, [pre-mRNA]: pre-mRNA concentration, [mRNA]: mature mRNA concentration. When comparing two samples, the parameters for the second sample are denoted with a prime (′) symbol. **c** The relationship between Δexon–Δintron and Δintron in the absence of differential decay rate, for various ratios of pre-mRNA abundance and Michaelis constant of RNA processing ($K_p$). Larger [pre-mRNA]/$K_p$ ratios are shown with *darker curves*. Positive and negative Δintron values correspond to transcriptional upregulation and downregulation, respectively (Methods). $V_{p,max}$ is assumed to be invariable. **d** The observed relationship between Δexon–Δintron and Δintron across 20 human tissues. Each data point represents the measurement for one gene in one tissue. The *yellow curve* denotes the trend line (average across sliding windows of 1000 data points). **e** The Pearson correlation coefficient between Δexon–Δintron and Δintron for genes with various read coverage. Each *circle* represents the set of genes that pass both the exonic and intronic read count cutoffs shown on the x axis and y axis (median read count across 20 tissues). The *circle* size represents the number of genes that pass the cutoffs, and the *color* shows the Pearson correlation between Δexon–Δintron and Δintron

RNA-seq data from a panel of 20 diverse human tissues[10] (data available at http://csg.lab.mcgill.ca/sup/pan_stability/), and observed widespread tissue-specific differences in the mRNA stability profiles (Supplementary Fig. 6). Comparison of these stability profiles with those obtained from a panel of mouse tissues suggests a high degree of conservation across species (Supplementary Fig. 7). Furthermore, genes that are involved in the same pathway are often co-regulated post-transcriptionally (Supplementary Fig. 8). In particular, pathway analysis of mRNA stability profiles revealed a large number of biological processes specifically upregulated or downregulated in the brain, suggesting a prominent role of post-transcriptional programs in shaping the brain transcriptome. This observation prompted us to take a closer look at regulation of mRNA stability in this tissue, since systematic analyses of mRNA stability programs in the brain are scarce, and it is not clear which factors contribute the most to the mRNA stability landscape of the brain.

We began by examining the potential binding sites of all human RBPs with available sequence preferences[19] as well as all conserved miRNAs[20], and used multiple linear regression to identify factors whose binding to 3′-untranslated region (UTR) of mRNA was predictive of mRNA stability in the brain (see Methods for more details). This analysis identified four miRNAs and two RBPs that were significantly predictive of brain mRNA stability (Fig. 3a). Specifically, presence of 3′ UTR binding sites for miR-124, miR-29, miR-9 and miR-128 was significantly associated with reduced mRNA stability, whereas binding sites of RBFOX and ZFP36 families of RBPs were significantly associated with increased stability.

Of the four miRNAs that were significant, three miRNAs, miR-124, miR-9, and miR-128, show highly specific expression in the brain, based on a dataset of miRNA expression profiles across 40 human tissue samples[21] (Fig. 3b), which is consistent with the observed brain-specific destabilization of their targets. Of these miRNAs, miR-124 and miR-9 are involved in development and function of the nervous system[22,23], and de-regulation of miR-128 is associated with tumors of the nervous system[24,25]. All four miRNAs have been previously shown to be able to decrease the abundance of their target transcripts[26–30], suggesting that they can indeed destabilize their targets. Also, of the two RBP families

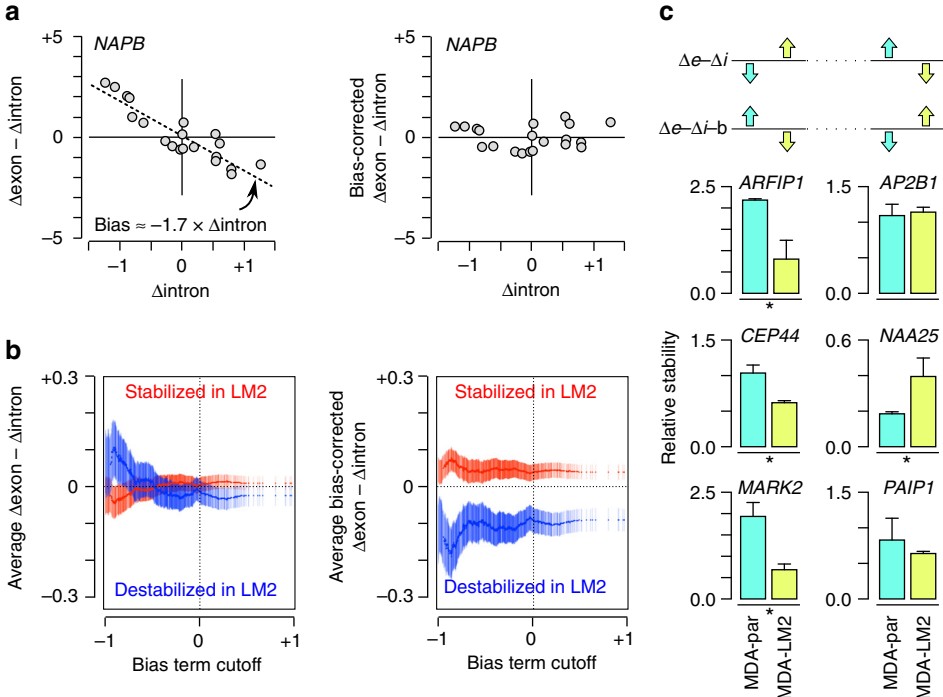

**Fig. 2** Removing bias from Δexon–Δintron improves inference of differential mRNA stability **a** Example demonstration of estimating and subtracting bias from Δexon–Δintron. We assume that the true differential mRNA decay rate is overall not correlated with the bias term, allowing us to estimate the bias term using regression. **b** Uncorrected (*left*) and bias-corrected (*right*) Δexon–Δintron for mRNAs that were previously reported to be stabilized (*red*) or destabilized (*blue*) in metastatic MDA-LM2 cells compared to parental MDA cells[17] (reported differential stability scores ≥ 0.95). Each point represents the set of genes that have a bias slope smaller than the corresponding cutoff on the *x* axis (points to the *left* correspond to more biased genes). The *y* axis corresponds to uncorrected or bias-corrected Δexon–Δintron for LM2 sub-line compared to parental MDA (positive means larger predicted stability in LM2). *Error bars* represent s.e.m. **c** Stability measurements for six mRNAs that were not previously detected as differentially regulated between MDA and LM2 cells. In each panel, the *y* axis represents the stability relative to 18 S rRNA. The *top panel* shows the predicted direction of change in stability based on uncorrected or bias-corrected Δexon–Δintron. Significant differences between the two cell types are marked with *asterisks* (*Mann–Whitney *U*-test on relative stabilities, *P* < 0.05). The *error bars* correspond to the s.d. (three biological replicates)

whose target transcripts were significantly up-regulated in brain, the RBFOX proteins show brain-specific expression (Fig. 3c), consistent with previous studies suggesting that RBFOX proteins stabilize their target mRNAs[19,31]. In contrast, ZFP36 destabilizes its targets by binding to their 3′ UTRs[32], in agreement with highly specific downregulation of ZFP36 gene and its paralog, ZFP36L1, in the brain (Fig. 3c). This suggests that ZFP36/ZFP36L1 proteins bind to the 3′ UTR of brain-specific mRNAs to destabilize them in non-neural tissues, resulting in highly specific expression of these genes primarily in brain where ZFP36/ZFP36L1 proteins are absent. The genes encoding for RBFOX and ZFP36 families of proteins themselves appear to be mostly regulated at the transcription level, based on analysis of intronic reads (Fig. 3d). Therefore, we propose a regulatory model in which brain-specific transcriptional activation/inhibition of these two RBPs establishes a post-transcriptional program that stabilizes brain-specific mRNAs (Fig. 3e). Together with rapid decay of non-neural transcripts by miRNAs, this parsimonious regulatory model, which consists of only six regulatory factors (Fig. 3e), explains ~10% of the observed variance of the mRNA stability profile of the brain, compared to ~31% of the variance that is reproducible across samples/platforms (Fig. 3f). Furthermore, this model is able to distinguish transcripts that show reproducible brain-specific stabilization from those that show destabilization, with an AUROC of 0.86 (area under the receiver operating characteristic curve, Supplementary Fig. 9).

**Dysregulation of RBFOX programs in Alzheimer's disease.** We combined the stability profiles of mRNAs in human brain with

the binding site predictions of the miRNAs and RBPs to reconstruct a high-confidence network of mRNA stability programs in brain (available at http://csg.lab.mcgill.ca/sup/pan_stability/). We considered a transcript to be a regulatory target of one of the six factors discussed in the previous section if that transcript had a putative binding site for that factor in its 3′ UTR, and the presence of the binding site was necessary to explain the stability of the transcript in the brain even after taking into account the effect of other stability factors (see Methods). This network overall encompasses 2499 interactions between six regulatory factors (four miRNAs and two RBPs) and 2138 transcripts (Fig. 4a). Several lines of evidence suggest that the edges in this regulatory network are more likely to represent bona-fide regulatory interactions than sequence-based binding site predictions alone. Specifically, high-confidence predicted RBFOX stability targets are 2.8-fold more likely to be bound by RBFOX proteins than transcripts that only have a match to the RBFOX motif in their 3′ UTR (Fisher's exact test $P < 4 \times 10^{-12}$, Fig. 4b), based on comparison to HITS-CLIP data of Rbfox1/2/3 in mouse brain[33] (see Methods). Similarly, ZFP36 high-confidence targets are more likely to be bound and downregulated by ectopically expressed ZFP36 in HEK293T cells[34] (Fisher's exact test $P < 0.05$, Fig. 4c). High-confidence predicted targets of miR-124 are also 3.6-fold more likely to be downregulated in HeLa cells that express an ectopic copy of miR-124[26], compared to transcripts that only have a match to miR-124 seed sequence (Fisher's exact test $P < 3 \times 10^{-6}$, Fig. 4d). Furthermore, our high-confidence network is significantly enriched for experimentally validated interactions that are collected from the literature for each of the four miRNAs

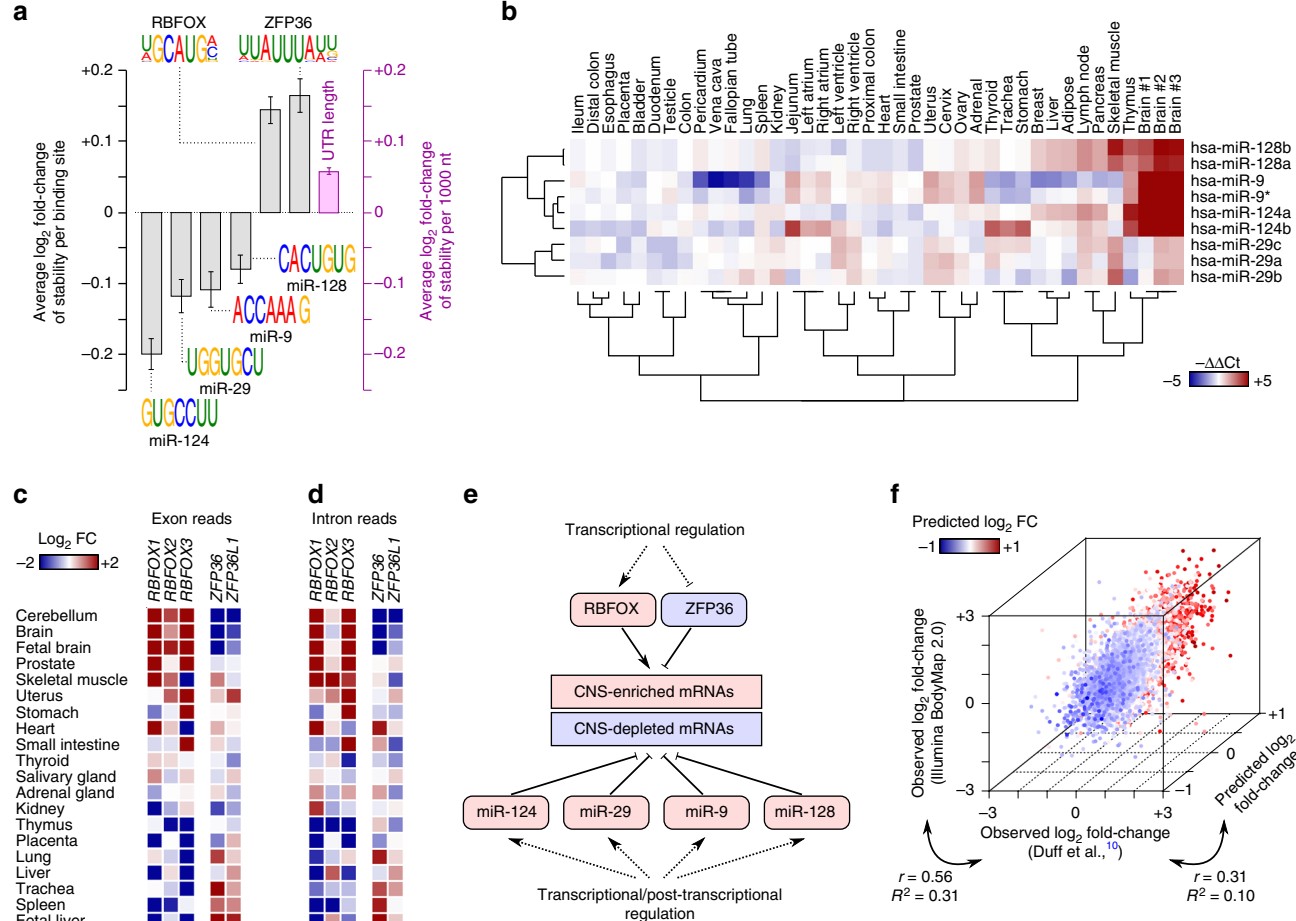

**Fig. 3** Factors that modulate mRNA stability in human brain. **a** Factors whose binding to the 3′ UTR is significantly associated with brain-specific mRNA stability are shown (FDR < 0.01, *t*-test of regression coefficients). The length of the 3′ UTR is also a significant predictor of mRNA stability, shown using the *right axis*. The *error bars* represent s.e.m. **b** Tissue-specific expression profiles of miRNAs that are associated with brain mRNA stability (data from ref. [21]). **c** Steady-state mRNA abundance of RBPs whose motif is associated with brain mRNA stability. **d** Transcriptional activity of RBPs, inferred from change in the abundance of intronic reads. **e** A schematic representation of the inferred mRNA stability model of human brain. **f** A 3D scatterplot of the brain mRNA stability profile based on RNA-seq data from ref. [10], RNA-seq data from Illumina BodyMap 2.0, and predictions based on presence of binding sites of miRNAs and RBPs (10-fold cross-validation). The latter is also represented by the color gradient. Each data point stands for one gene

miR-124, miR-128, miR-29, and miR-9[35] (Fig. 4e). The high-confidence 3′UTR binding sites of all these four miRNAs are on average significantly more conserved than their adjacent sequences, including the binding sites that were not previously validated in the literature (Supplementary Fig. 10a). Interestingly, the intersection of our network with the predictions of TargetScan[20], which uses conservation to identify miRNA targets, shows an even larger enrichment of functional miRNA-target interactions compared to either method alone (Supplementary Fig. 10b, c), suggesting that our method provides orthogonal information compared to conservation-based approaches.

We observed significant enrichment of several pathways related to neuronal function among the stability targets of RBPs in our high-confidence network (Supplementary Fig. 11). Of particular note, we found that the RBFOX network is most highly enriched for genes that are involved in synaptic transmission (2.4-fold enrichment, Fisher's exact test $P < 1 \times 10^{-9}$). Synaptic transmission is the main pathway that is affected in Alzheimer's disease (AD)[36], suggesting that a defect in the RBFOX stability program could lead to de-regulation of synaptic genes and, subsequently, result in AD or AD-like phenotypes. To test this hypothesis, we analyzed the RNA-seq data from the dorsolateral prefrontal cortex of six patients with advanced AD as well as five control

subjects[37] (see Methods), using our computational framework for obtaining unbiased estimates of mRNA stability. Interestingly, we found that the average mRNA stability profile of AD is significantly anti-correlated with the brain stability signature that we obtained from analysis of the panel of 20 tissues ($r = -0.13$, $P < 2 \times 10^{-35}$, Supplementary Fig. 12a), indicating that brain-specific transcripts are, on average, destabilized in advanced AD. The pathway that showed the most significant enrichment among the top destabilized genes was synaptic transmission (Supplementary Fig. 12b), supporting the hypothesis that destabilization of synaptic transmission genes may be associated with AD. Indeed, the stability targets of RBFOX showed a large enrichment among genes that are destabilized in the average AD brain (Mann–Whitney $U$-test $P < 3 \times 10^{-10}$, Fig. 5a and Supplementary Fig. 13a). Given that many of RBFOX targets are specifically expressed in neurons, one possibility is that downregulation of these transcripts reflects neuronal loss, which is commonly seen in advanced AD[36]. However, we observed that RBFOX targets were significantly more downregulated compared to other neuron-specific genes (Supplementary Fig. 13b). In addition, RBFOX targets whose expression is not limited to neural cells were also downregulated in AD (Supplementary Fig. 13c)—a change in the ratio of neural cells would have a relatively small

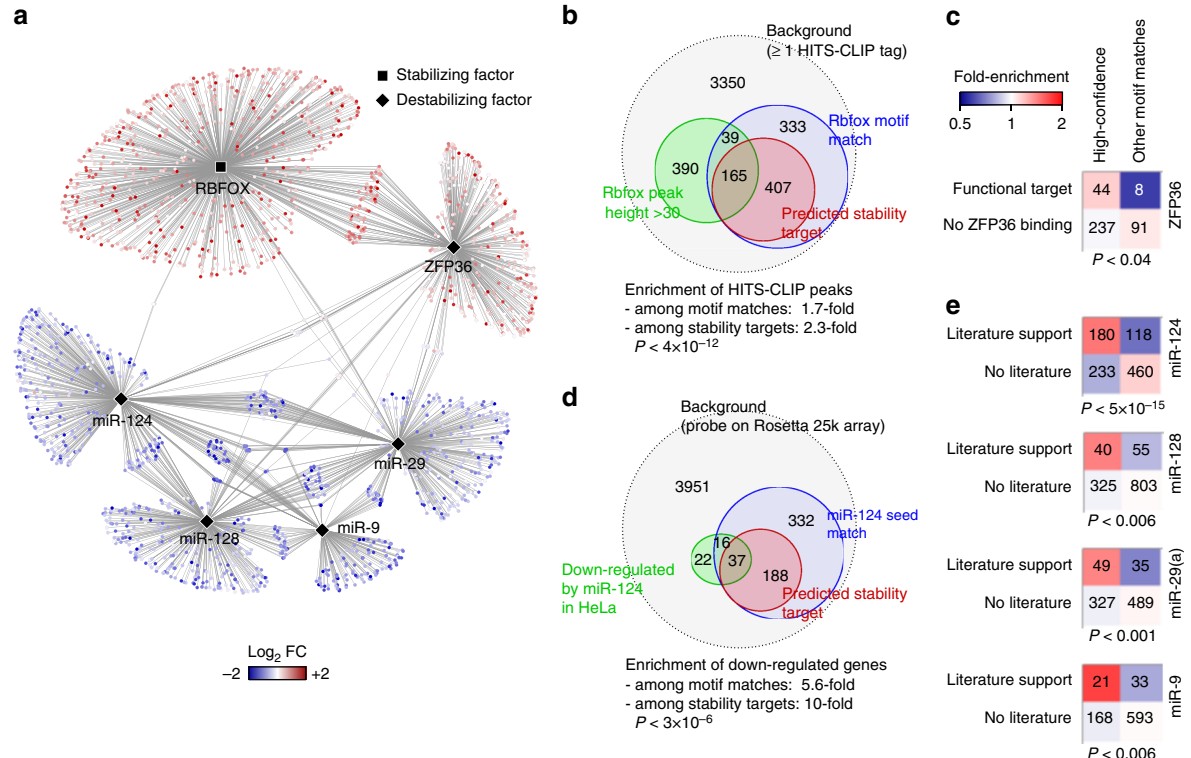

**Fig. 4** A high-confidence network of brain mRNA stability. **a** The high-confidence network of RBPs, miRNAs and their targets in human brain. Node color represents the mRNA stability in brain relative to average tissue. **b** Venn diagram of the genes that contain a match to the RBFOX motif in their 3′ UTR, the subset that is in the high-confidence network (referred to as "stability targets"), and the genes whose orthologs in mouse are bound by Rbfox1/2/3 proteins[33]. Only human genes that have a mouse ortholog are included in the analysis. **c** Overlap of high-confidence ZFP36 targets with experimentally identified functional targets of ZFP36[34]. The color gradient represents fold-enrichment relative to the extent of overlap that would be expected by chance. Functional targets were defined as transcripts with at least one ZFP36 PAR-CLIP cluster in their 3′ UTRs that were > 2-fold downregulated after ectopic expression of ZFP36 in HEK293T cells. The bottom row represents transcripts without a significant PAR-CLIP cluster in 3′ UTR. **d** Venn diagram of genes with a match to miR-124 seed sequence, the subset in the high-confidence network, and genes that are downregulated when miR-124 is expressed in HeLa cells[26]. **e** Enrichment of experimentally validated targets of miRNAs among our predicted high-confidence stability targets. The miRNA targets that are supported by literature are obtained from miRTarBase[35]. The color gradient is similar to **c**. All P values are based on Fisher's exact test

impact on the apparent abundance of these transcripts since they are also expressed in other cell types, suggesting that the observed destabilization of RBFOX targets is not an artifact of neuronal loss in AD. These observations are not data-set-specific, and we were able to replicate them in an independent cohort of four AD patients and four control subjects[38] (Supplementary Fig. 14). In addition, the intersection of RBFOX stability target set and the AD-destabilized gene set is > 2.1-fold enriched for the synaptic transmission pathway compared to either of the RBFOX target set or AD-destabilized gene set alone (Fisher's exact test P < 0.03, Fig. 5b). These results suggest that RBFOX-bound transcripts that encode synaptic transmission proteins are destabilized in AD.

Among the three RBFOX proteins, we found that the transcription of *RBFOX1* gene is significantly reduced in the AD brain compared to control samples (two-tailed t-test P < 0.008, Supplementary Fig. 15a). This observation is supported by analysis of a larger dataset of microarray measurements[39], which shows ~ 2-fold decrease in the abundance of *RBFOX1* mRNA in AD subjects compared to normal individuals (two-tailed t-test P < 3 × 10^{-53}, Fig. 5c and Supplementary Fig. 15b). In contrast, other neuron-specific genes showed only a median downregulation of 1.2-fold in AD (Supplementary Fig. 15b). This suggests that RBFOX1 downregulation may contribute to the deregulation of the RBFOX stability program in AD. Indeed, we observed that RNAi-mediated knockdown of *RBFOX1* in differentiated primary human neural progenitor cells[40] leads to a

transcriptome shift that is significantly correlated with the AD stability signature ($r = 0.13$, $P < 4 \times 10^{-33}$, Fig. 5d). Furthermore, the predicted RBFOX stability targets are most highly enriched among transcripts that are both destabilized in AD and down-regulated by RBFOX1 knockdown (Fig. 5d). This can also be observed in mouse neurons, where knockdown of Rbfox1/3 leads to downregulation of the orthologs of AD-destabilized RBFOX targets, and ectopic expression of a cytoplasmic form of Rbfox1 rescues the expression of these genes (Supplementary Fig. 16).

## Discussion

By inferring mRNA half-life from RNA-seq data, we were able to obtain a global view of the mRNA stability landscape of human tissues, which revealed a prominent role of mRNA stability in shaping the transcriptome of the brain. By combining the brain mRNA stability signature with the consensus sequence binding preferences of miRNAs and RBPs, we identified four miRNAs and two RBPs as the primary determinants of mRNA stability in the brain. It is, however, important to note that the consensus binding preferences of RBPs and miRNAs are often poor predictors of their in vivo binding sites, and spurious matches to these consensus sequences are abundant. Among factors that contribute to such false positive hits are the RNA structure[41] and the inaccuracies in known consensus binding sequences[42]. Our high-confidence stability network (Fig. 4a) reduces these false positives by identifying "functional" targets of each factor, i.e.

mRNAs that not only have a sequence match to its consensus binding sequence, but also have a stability that is consistent with the effect of binding of that factor. Interestingly, even though we did not use structural information to construct our high-confidence network, the expected structural preferences of RBPs are reflected in this network, with high-confidence RBFOX1 and ZFP36 binding sites showing higher local accessibility compared to spurious sequence matches (Supplementary Fig. 17a, b). Furthermore, high-confidence binding sites of miR-124 contain conserved sequences that are compatible with extended base-pairing beyond the miR-124 seed region, in contrast to spurious matches to miR-124 recognition sequence (Supplementary Fig. 17c). These observations suggest that by combining mRNA stability with consensus binding preferences, we can overcome

some of the challenges in computational identification of the true binding sites of RBPs and miRNAs.

Among the factors that we studied in brain, RBFOX proteins are primarily known for their role in regulating alternative splicing[43], and more recently as potential regulators of mRNA stability[19,31]. Surprisingly, we found that RBFOX proteins regulate transcripts that are functionally relevant to development of Alzheimer's disease (AD) and are destabilized in brains of AD patients. Intriguingly, the *RBFOX1* gene itself is downregulated in AD, consistent with previous studies that have found an association between rare heterozygous deletions overlapping the *RBFOX1* locus and early-onset familial AD[44]. Furthermore, we found that RNAi-mediated knockdown of *RBFOX1* is able to partially recreate the AD stability signature in human neural

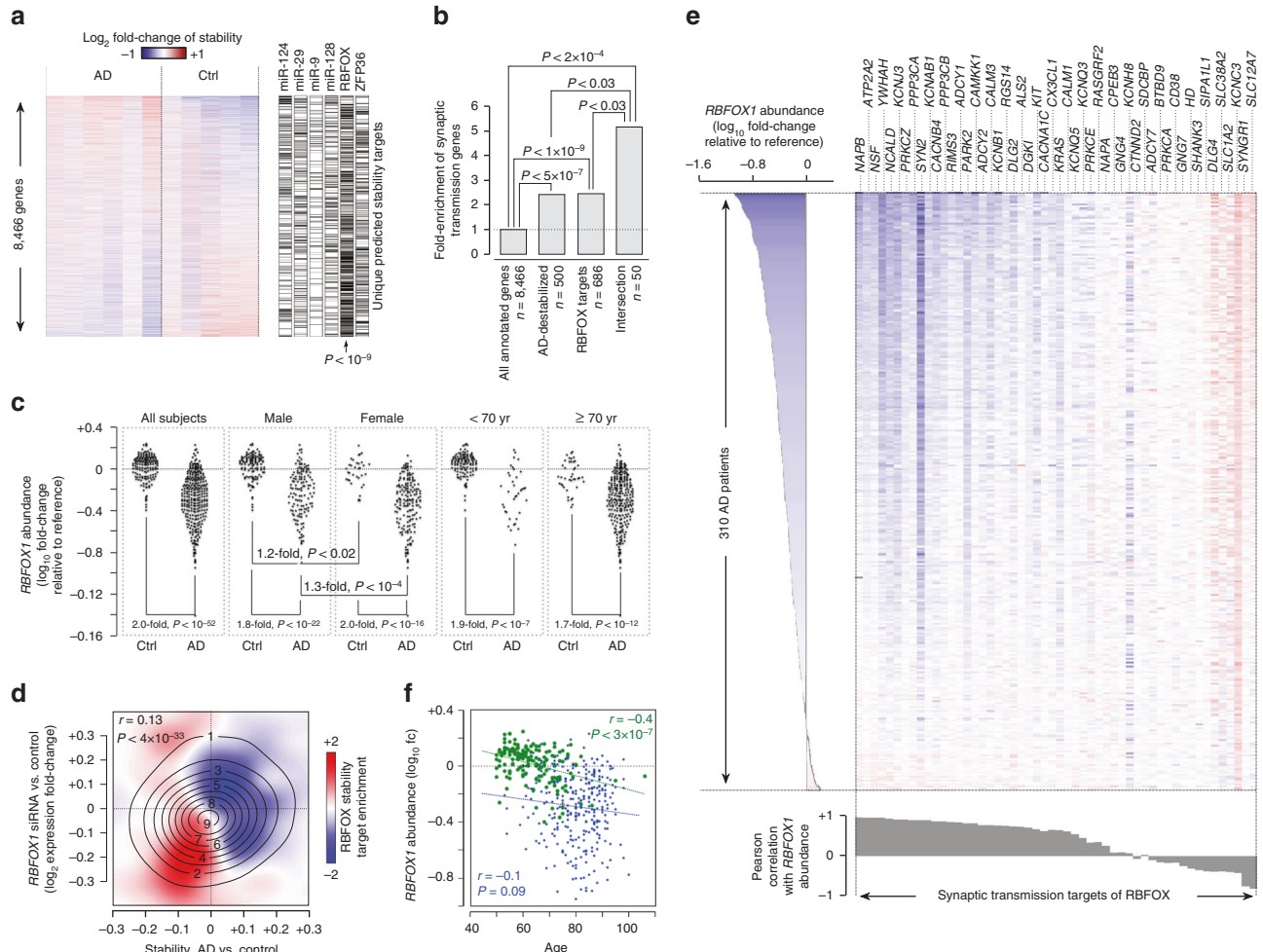

**Fig. 5** De-regulation of RBFOX stability programs in Alzheimer's disease. **a** Enrichment of targets of brain stability factors among transcripts that are stabilized or destabilized in Alzheimer's disease (AD). The *heat map* shows the stability of transcripts relative to average across all samples. Genes are sorted by t-score of difference between the AD ($n = 6$) and control (Ctrl, $n = 5$) groups. The unique targets of each factor (i.e., transcripts that are not targeted by any other factor) are shown on the *right*. The P value of enrichment of RBFOX targets among AD-destabilized genes is shown on the bottom right (Mann–Whitney U test). **b** Enrichment of synaptic transmission genes among the top 500 transcripts destabilized in AD, the high-confidence RBFOX stability targets, and the intersection of the two sets. **c** Comparison of abundance of RBFOX1 mRNA in the brain of AD and non-demented (Ctrl) individuals. The *left panel* compares the RBFOX1 mRNA abundance between 310 AD and 157 non-demented individuals (average of two RBFOX1 microarray probes)[39], whereas other *panels* represent subgroups based on gender or age. P values are based on two-tailed Student's t-test. **d** Correlation of AD-associated change in stability (*x* axis) with the change in expression after RBFOX1 knockdown (*y* axis, data from ref. [40]). The contours represent the probability density function for all genes, and the *color gradient* represents the density of genes that are RBFOX stability targets minus density of other genes (*red*: higher density of RBFOX targets). **e** Expression of synaptic transmission genes that are regulated by RBFOX proteins across 310 AD patients[39]. Each row represents one individual, sorted by the ascending order of RBFOX1 expression (shown on the *left*). Each *column* represents one gene, sorted by the descending order of correlation with RBFOX1 expression (shown at the *bottom*). **f** Scatterplot of RBFOX1 abundance vs. age in 310 AD patients (*blue dots*, $r = -0.1$) and 157 non-demented individuals (*green dots*, $r = -0.4$)[39]. Overall Pearson correlation is $-0.46$ ($P < 10^{-26}$)

progenitor cells, suggesting a role of RBFOX1 in AD-associated changes in neural cells.

An almost-invariable feature of Alzheimer's disease is synaptic impairment, which is often associated with aggregation of β-amyloids[45]. RBFOX1 is previously reported to regulate the splicing of amyloid precursor protein (APP), with de-regulation of this splicing program leading to an APP isoform that contributes to β-amyloid aggregation[46]. Here, our analyses suggest an alternative association between RBFOX1 and AD, where down-regulation of RBFOX1 may directly affect the stability and abundance of mRNAs that encode synaptic transmission proteins. If this is the case, then we should expect a correlation between the extent of *RBFOX1* downregulation and the severity of de-regulation of synaptic transmission pathway in AD patients. Indeed, there is a strong correlation between *RBFOX1* expression and abundance of RBFOX-regulated synaptic genes across 310 AD individuals (median $r = 0.62$, two sided $t$-test $P < 3 \times 10^{-6}$, Fig. 5e).

These findings provide new insights into pathways that may lead to increased risk of development of AD. For example, while the primary risk factors of AD are age and gender[47], the mechanisms through which these factors affect the AD-susceptibility are largely unknown. Surprisingly, we found that *RBFOX1* expression is highly correlated with gender in both normal and AD individuals, and with age in normal individuals. Specifically, *RBFOX1* is expressed at lower levels in females (Fig. 5c) and older subjects (Figs. 5c, f), consistent with higher risk of AD in these individuals. These associations, however, can only partially explain the large magnitude of differences that are seen between AD and healthy individuals (Figs. 5c, f), consistent with the notion that AD is a complex and multi-factor disease[48]. Furthermore, interpretation of AD transcriptomics data is particularly challenging given the confounding effect of neuronal loss on gene expression measurements, despite our attempts to control for this confounding factor in our analyses (Supplementary Figs. 13 and 15).

Together with the functional relevance of RBFOX1 stability targets, and the similarity of the expression profile of RBFOX1-deficient cells with the stability profile of AD brain, these observations suggest that defects in RBFOX1 stability program may contribute to the loss of synaptic function in AD. We note that, based on our analysis of tissue-specific RNA-seq data, mRNA stability plays a pivotal role in shaping the transcriptome and the landscape of active biological processes in many tissues (Supplementary Figs. 6–8), and therefore de-regulation of programs that modulate mRNA stability are likely to lead to various diseases beyond those of the nervous system. Our method for obtaining unbiased estimates of mRNA stability can be applied to a wide range of diseases with available RNA-seq data to reveal disease-associated stability programs.

## Methods

**RNA-seq and microarray data**. For analysis of human tissues, RNA-seq data of 20 human tissues were obtained from ref. [10] (SRA accession SRP056969). For analysis of AD-associated stability programs, RNA-seq data from brain tissue of nine AD and eight control individuals were obtained from ref. [37] (GEO accession GSE53697). For analysis of MDA-parental and MDA-LM2 cells, RNA-seq data from ref. [15] were used (GEO accession GSE45162). Reads were mapped to the hg19 assembly of human genome using TopHat2[49] (or HISAT2[50] in case of MDA/LM2 data) with default parameters. Intronic and exonic coordinates of genes were extracted from GENCODE v19[51], and gene-level read counts were calculated for introns and exons separately using HTSeq[52], including only reads with MAPQ score ≥ 30. Three AD and three control individuals from ref. [37] were excluded, as analysis of their RNA-seq data resulted in a disproportionately large number of genes with zero mapped reads and an unusual clustering of these samples (Supplementary Fig. 18), possibly due to low sequencing read quality.

Mouse tissue RNA-seq data were obtained from ref. [53] (GEO accession GSE29278), and were mapped to mm10 assembly using HISAT2[50]. Gene-level

intronic and exonic read counts were obtained as above, using Ensembl[54] release 87 annotations.

Microarray data from 310 AD and 157 control individuals were obtained from ref. [39]. Gene-level abundances in each sample were obtained by averaging the relative intensities of probes that mapped to the same gene (logarithm of ratio of probe intensity between the two channels, with one channel representing pooled RNA reference sample).

Gene-level exonic and intronic read counts for Illumina BodyMap 2.0, ENCODE RNA-seq data for cell lines HMEC, NHEK, HUVEC, HSMM, NHLF, H1 hESC, HepG2, GM12878, and K562, as well as RNA-seq datasets presented in Supplementary Figs. 2b and 5 were obtained from ref. [5]. Other RNA-seq and microarray measurements used in this work were taken directly from the supplementary data of their respective publications.

**Modeling the pre-mRNA and mRNA abundance**. The model of RNA metabolism we used includes three main steps (Fig. 1a). For each gene, DNA is first transcribed to produce pre-mRNA at rate $V_t$, following zero-order production kinetics with rate constant $k_t$. Then, the pre-mRNA is processed into mature mRNA at rate $V_p$ following Michaelis–Menten kinetics, with parameters $V_{p,max}$ as the maximum processing rate obtained at saturating levels of the pre-mRNA, and Michaelis constant $K_p$ representing the pre-mRNA concentration at which $V_p$ is equal to half of $V_{p,max}$. Michaelis-Menten is one of the simplest approaches to model kinetics of saturation, and is applied widely to enzymatic reactions, including RNA processing[9]. The mature mRNA is degraded at rate $V_d$, which follows first-order elimination kinetics[5], with a decay rate constant of $k_d$. At steady state, the input and output of the system must be equal, and therefore the rate of transcription is equal to the mRNA decay rate[5], as well as the rate of mRNA processing.

Subsequently, we modeled the ratio of abundance of each of pre-mRNA and mature mRNA between any two samples $s$ and $s'$ as a function of $k_t$ and $k'_t$, representing the transcription rate in the two samples, respectively, and $k_d$ and $k'_d$, representing the decay rate constant in the two samples, respectively (Fig. 1b and Supplementary Fig. 1). Values of $V_{p,max}$ and $K_p$ were assumed to remain constant between two samples for simplification. We note that these parameters mainly depend on the availability of the RNA processing machinery, which is ubiquitously expressed across most cells[55,56], and the intrinsic properties of the pre-mRNA/processing machinery complex. Therefore, the assumption of invariability of $V_{p,max}$ and $K_p$ for each gene across tissues/samples is reasonable in most cases. Together with the assumption of steady sate, this model results in a set of equations that are solved to identify the relationship among the abundance of pre-mRNA, the abundance of mature mRNA, and the differential rates of transcription and decay between two samples, as shown in Fig. 1b and Supplementary Fig. 1. We note that solving the kinetic equations in the absence of the above simplifying assumptions (Supplementary Fig. 4a) suggests that when the variation in $V_{p,max}$ is considerably smaller in magnitude than the variation in degradation rate, its effect is negligible and can be ignored. The same is true for $K_p$ when its variation is considerably smaller than the variation in the concentration of pre-mRNA. Supplementary Fig. 4b also shows that even when $V_{p,max}$ and $K_p$ are highly variable, the method presented in this paper still captures the change in mRNA stability better than uncorrected Δexon–Δintron.

**Removing bias from Δexon–Δintron**. We took the average of all samples within each dataset as the reference sample $s_{ref}$, and then for each gene in each tissue/sample $s$, we calculated gene-level Δexon and Δintron values by comparing to $s_{ref}$ using DESeq. Specifically, for each of the exonic and intronic read sets separately, we used variance-stabilized transformation from DESeq[57] to estimate the logarithm of read abundances for each gene, and then subtracted the average value across the samples to obtain Δexon and Δintron.

To filter against genes with low read counts that lead to noisy estimates of Δexon and Δintron, we determined a minimum read cutoff $\theta$ that nearly maximizes the correlation between Δexon and Δintron. Specifically, we determined the value of read count cutoff that maximizes the correlation ($r_{max}$) between Δexon and Δintron, and then selected the minimum $\theta$ that results in $r \geq 0.99 \times r_{max}$ for genes with both exonic and intronic read counts ≥ $\theta$.

To estimate the logarithm of ratio of $k_{d,ref}$ and $k_d$ (corresponding to decay rates in the reference sample $s_{ref}$ and sample $s$, respectively) for this selected subset of genes, we estimated the bias term shown in Fig. 1b for each gene separately. This bias term depends on the RNA processing Michaelis constant $K_p$, and the concentration of pre-mRNA in query sample $s$ and reference sample $s_{ref}$, shown below as [pre-mRNA] and [pre-mRNA]$_{ref}$, respectively:

$$f_{bias}(\Delta i) = \log_2 \frac{K_p + [\text{pre-mRNA}]_{ref}}{K_p + [\text{pre-mRNA}]}$$

$$= \log_2 \frac{\frac{K_p}{[\text{pre-mRNA}]_{ref}} + 1}{\frac{K_p}{[\text{pre-mRNA}]_{ref}} + \frac{[\text{pre-mRNA}]}{[\text{pre-mRNA}]_{ref}}}$$

$$= \log_2 \frac{c+1}{c+2^{\Delta i}}$$

Here, $f_{bias}$ is the gene-specific bias function, and $\Delta i$ represents Δintron, i.e., the logarithm of change in abundance of intronic fragments relative to the reference sample. The parameter $c$ is an unknown gene-specific constant that depends on the ratio of $K_p$ and [pre-mRNA]$_{ref}$. As shown in Fig. 1c, $f_{bias}$ can be approximated as a

linear function of $\Delta i$ over a wide range of values of the constant $c$. Therefore, we estimated the function $f_{bias}$ for each gene by least-square linear regression of $\Delta$exon–$\Delta$intron vs. $\Delta$intron across all tissues/samples in each dataset, assuming that the change in mRNA decay constant across samples is independent of the bias term, and therefore the partial regression coefficient in the model $\Delta e - \Delta i = \Delta \log (k_d) + f_{bias}(\Delta i)$ is equal to the regression coefficient in the model $\Delta e - \Delta i \sim f_{bias}(\Delta i)$. The value of the bias function in each sample is then subtracted from $\Delta$exon–$\Delta$intron to obtain an unbiased estimate of $\log_2(k_{d,ref}/k_d)$, which is equal to differential stability. These steps are implemented in a software package available at https://github.com/csglab/REMBRANDTS. The mRNA stability estimates presented in this paper are available at http://csg.lab.mcgill.ca/sup/pan_stability/.

**Gene selection for qRT-PCR validation of stability estimates.** We used RNA-seq data from ref. [15] to measure uncorrected and bias-corrected $\Delta$exon–$\Delta$intron, and then identified genes that were inferred to be up-regulated or downregulated in MDA-parental or MDA-LM2 cells based on each measure (two-tailed Student's $t$-test $P < 0.05$). We then excluded all genes with previously reported differential stability scores > 0.5 or < –0.5 between MDA-parental and MDA-LM2 cells[17], and among the remaining genes, identified those that had conflicting predictions from uncorrected and bias-corrected $\Delta$exon–$\Delta$intron. We sorted these genes based on their bias slopes, and selected the three most highly biased genes that were predicted by uncorrected $\Delta$exon–$\Delta$intron to be stabilized in MDA-LM2, as well as the three most highly biased genes that were predicted based on uncorrected $\Delta$exon–$\Delta$intron to be destabilized in MDA-LM2 relative to MDA-parental—all six genes had the reverse prediction based on bias-corrected $\Delta$exon–$\Delta$intron. We then measured the stability of these genes by $\alpha$-amanitin inhibition of transcription followed by qRT-PCR, as described below.

**Sample preparation for qRT-PCR.** Poorly metastatic MDA-parental (MDA-MB-231) and highly metastatic MDA-LM2 cells[16] were obtained from Tavazoie lab (The Rockefeller University), and were tested to ensure absence of mycoplasma contamination. Cells were seeded at $2 \times 10^5$ per well in six-well plates (in biological triplicate). The following day, RNA was extracted from three MDA-parental and three MDA-LM2 wells for the 0-h time point. The remaining wells were treated with 10 μg/mL $\alpha$-amanitin (Sigma) for 9 h before RNA extraction.

**Validation of mRNA stability estimates by qRT-PCR.** RNA was isolated from MDA-parental and MDA-LM2 cells using a total RNA isolation kit with on-column DNase treatment (Norgen). Upon first-strand cDNA synthesis (SSIII, Life Technologies), relative levels of each mRNA of interest were assessed by qRT-PCR (ABI 7300 Real-Time System), using 18S as the endogenous control. Primer sequences are listed in Supplementary Table 1. For each cell-line, relative stability for each gene was defined as the 9-hr/0-hr ratio. Statistical significance was determined using Mann–Whitney $U$-test on relative stabilities.

**Analysis of RBP and miRNA binding sites.** We limited the analysis of RBP and miRNA binding sites to genes for which all isoforms had the same 3′ UTR coordinates, and the 3′ UTR was composed of a single exon, in order to minimize the possibility of confounding effects of alternative splicing. The 3′ UTR coordinates were extracted from GENCODE v19[51].

For analysis of RBP binding sites, we first collected a non-redundant compendium of available sequence preferences for human RBPs. We obtained 175 position frequency matrices (PFMs) representing 99 human RBPs from CisBP-RNA[19], including PFMs with direct experimental evidence and those inferred by homology. Then, we calculated all pairwise PFM similarities using MoSBAT[58], and then used affinity propagation[59] to cluster the PFMs based on similarity, keeping only the "exemplar" from each cluster. This reduced the total number of PFMs to 126, which we call the non-redundant RBP motif set. Then, we scanned the 3′ UTR sequences with each PFM using AffiMx from the MoSBAT package[58], resulting in a vector of PFM scores that represents the affinity of the corresponding RBP for binding to different 3′ UTRs.

For analysis of miRNAs, we obtained the seed sequences of 153 conserved human miRNA families from TargetScan[20], and searched for exact matches to each seed sequence within 3′ UTRs. For each seed sequence, the number of matches within each 3′ UTR was recorded, resulting in a vector of miRNA seed match counts across the 3′ UTRs.

To identify RBPs and miRNAs that are associated with brain-specific stability profiles, we used the unbiased gene-level stability measures as the response variable in multiple linear regression, with RBP affinities and miRNA seed match counts as predictor variables. We also included 3′ UTR length, nucleotide frequencies, and dinucleotide frequencies as additional predictor variables in regression, in order to control for the confounding effect of these variables. RBPs and miRNAs whose binding sites were significantly associated with brain-specific stability were identified based on $t$-test of regression coefficients at FDR < 0.01.

**High-confidence stability network of brain.** In order to identify high-confidence stability targets of each of the two RBPs and four miRNAs in the brain, we searched

for genes that had a 3′ UTR binding site for that factor and for which that factor was necessary to explain the differential stability, even after taking into account all other factors. Specifically, for each factor $f$, we first repeated the multiple linear regression of the previous section after excluding $f$ from the set of predictor variables, and took the residual of regression as the differential stability that remains unexplained after considering all factors except $f$. Then, we sorted the genes by the ascending order of this residual value, and at each residual value $x$, we calculated the distance ($D$) of the cumulative distribution functions for genes that had a binding site for $f$ (the gene set $F$) and genes that did not have a binding site for $f$:

$$D(x) = |P(X \leq x | g \in F) - P(X \leq x | g \notin F)|$$

The $x$ that maximized $D$ was taken as the residual cutoff that marks stability targets of $f$. Therefore, genes for which the residual value was $\geq x$ (for RBPs) or $\leq x$ (for miRNAs) and had a binding site for $f$ were taken as high-confidence stability targets of $f$. Note that the maximum value of $D$ is equal to the two-sample Kolmogorov–Smirnov statistic for the distributions of the residual values between genes that have a binding site for $f$ and genes that do not have a binding site for $f$. We note that this procedure is similar to the "leading-edge" analysis described previously for GSEA[60], except that GSEA uses a weighted Kolmogorov–Smirnov statistic instead of the unweighted statistic that we used. The high-confidence stability network of brain is available at http://csg.lab.mcgill.ca/sup/pan_stability/.

For evaluation of the high-confidence stability targets of RBFOX proteins, we obtained potential RBFOX1/2/3 targets by identifying human orthologs of mouse genes that have at least one Rbfox1/2/3 binding site in their 3′ UTRs based on HITS-CLIP[33]. We used mouse genes that had at least one reported Rbfox1/2/3 peak with height $\geq 30$, and identified their human orthologs using Ensembl[54], taking only genes with one-to-one orthology. As the background, we used human genes whose mouse orthologs had at least one HITS-CLIP tag, in order to remove non-expressed genes.

For validation of predicted ZFP36 stability targets, we obtained PAR-CLIP data as well as RNA-seq measurements for ectopic expression of ZFP36 in HEK293T cells from ref. [34]. Functional ZFP36 targets were defined as genes with at least one PAR-CLIP cluster in their 3′ UTRs that also showed $\geq 2$-fold downregulation in ZFP36-expressing cells compared to control.

For validation of miRNA targets, we obtained experimentally validated targets of hsa-miR-124-3p, hsa-miR-128-3p, hsa-miR-29(a/b/c)-3p, and hsa-miR-9-5p from miRTarBase[35] release 6.1, which is a database of miRNA-target interactions collected from literature. For each miRNA, we removed target genes with ambiguous miRNA-target interactions (classified as both functional and nonfunctional in miRTarBase). Fisher's exact test was used to determine the significance of enrichment of functional miRTarBase miRNA-target interactions in our high confidence network, relative to miRNA seed sequence matches.

We also obtained the list of genes that are downregulated after ectopic expression of miR-124 in HeLa cells from ref. [26]. We used the set of genes with at least one probe on Rosetta 25k array as the background, in order to control for unobserved gene expression values, and calculated the enrichment of high-confidence stability targets of miR-124 among downregulated genes using Fisher's exact test.

**Code availability.** The scripts used for the analyses are available from https://github.com/csglab/REMBRANDTS.

**Data availability.** The RNA-seq and microarray data referenced in this study are available from GEO (https://www.ncbi.nlm.nih.gov/geo/) and SRA (https://www.ncbi.nlm.nih.gov/sra), using accession numbers GSE53697[37], GSE45162[15], GSE29278, and SRP056969[10]. Processed data are available from http://csg.lab.mcgill.ca/sup/pan_stability/.

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

## Acknowledgements

We thank Naghmeh Nikpoor (McGill University) for contribution to the processing of RNA-seq data. This work was supported by funds from McGill University Faculty of Medicine to H.S.N., and National Cancer Institute (NCI) grant CA194077 to H.G.

## Author contributions

H.S.N. conceived and designed the study, and developed the computational methods. R.A. and H.S.N. analyzed and interpreted the data. H.G. designed the experiments, and L.F. and H.G. performed the experiments. H.S.N. wrote the manuscript with contribution from R.A.

## Additional information

**Competing interests:** The authors declare no competing financial interests.

