## [Peer Review File · Nature Communications]

Reviewers' comments:

Reviewer #1 (Remarks to the Author):

In this manuscript, Najafabadi proposed a novel method to infer changes in mRNA decay rate from RNA-Seq data by separate analysis of exonic vs. intronic reads. This method is an extension of a previous method published by Gaidatzis et al., based on a kinetic model relating transcription, pre-RNA splicing and RNA delay. Using this method, the author inferred mRNA stability in human tissues including the brain, which was then correlated with binding sites of RBPs and miRNAs. Interestingly, two RBPs (RBFOX1 and ZFP36) and 4 miRNAs were identified as major regulators. In addition, the author also found that RBFOX1 is downregulated in AD brains, which is correlated with destabilization of RBFOX targets. As a method paper, the idea is interesting and the results appear to be promising, but it is currently limited by analysis of single dataset, so the advantage and robustness of the method are not fully demonstrated. More caution is required to interpret the observations from AD brains.

Major concerns:

Since this work aims to improve a previous method, a critical point is to demonstrate that the new method with bias correction outperforms the original method. In this regard, the authors made an interesting observation on the negative correlation between $\Delta E - \Delta I$ vs ΔI . The kinetic model appears to make sense, despite some assumptions and simplification not clearly justified. The inference of the gene-specific bias is based on the assumption that there is no change in degradation, which does not seem to be reasonable (Fig. 1c,d). The author only mentioned this in the Fig. 1c legend. This should really be clearly described in Method and justified.

The only direct comparison with the previous method was presented in Fig. S3 and the author claimed a 32% improvement in Pearson correlation (from ~ 0.25 to ~ 0.34) between predicted and experimentally measured differential stability. This reviewer noticed that the correlation reported by Gaidatzis et al. is 0.29, and it is unclear why is the difference, which will favor the authors' claim. Also, it will be helpful to show the correlation of mRNA stability with and without bias correction so that readers can get an intuitive idea how the correction affects the results.

Given the critical importance to demonstrate the advantage of the revised method, this reviewer feels it is essential to show the improvement is robust. While I am not aware whether there are other datasets with both RNA-Seq and mRNA stability measurements for the same set of samples, additional direct comparison would be very helpful.

It will also be important to analyze an independent RNA-Seq dataset, e.g., derived from mouse tissues, to see whether the bias shown in Fig. 1d, as well as estimates of gene-specific stability in the same tissues, is reproducible. The use of mouse tissues is also handy to avoid complication of RNA quality issues inherent in post-mortem human samples.

I have more reservations on the interpretation of the results from AD brains. I think the result presented in this paper is at best correlative, so claims such as "RBFOX-mediated stabilization of transcripts that encode synaptic transmission proteins is a critical factor for proper functioning of the brain, and defects in this regulatory program can contribute to the onset or progression of AD." should be avoided. An alternative interpretation of the results is the neuronal loss in advanced AD patients, since RBFOX1 and many transcripts identified by the author as RBFOX1 targets are neuron-specific.

The age/gender difference of RBFOX1 expression in normal brains is not convincing (Fig. 4c,f) given the magnitude of the difference and the potential complications such as RNA quality.

The siRNA KD data used in Fig. 4d is not reliable, since RBFOX1 expression is very low even in

control samples. This is likely due to poor differentiation of the cells. The poor quality of this dataset is also clear from Fig. 2 of the original Fogel paper, since the shGFP samples clusters together with shRBFox1, instead of WT. This dataset should not be used to avoid misleading readers and contamination of the literature.

Minor issues:

There appears to be some inconsistency in reference numbers. For example, Ref. 9 is not the paper with 20 human tissues.

Fig. 4f, green dots in the figure, but red dots in the legend.

Online Methods, additional explanation of the kinetic model will be helpful (e.g., why is the addition of $V_t=V_p=V_d$? What choose Michaelis-Menten kinetics to model splicing?). The author might do a better job to put the model into similar work in the literature?

Fig. S1 legend, "...results in an apparent anti-correlation between transcriptional and post-transcriptional regulation of pathways". This is certainly not true for all panels.

Related to Fig. 4c,e and Fig. S9, it will be helpful to show the level of RBFox2/3 from microarray data as well, and show the abundance of RBFox1/2/3 estimated from exonic reads in RNA-Seq data.

Reviewer #2 (Remarks to the Author):

This paper describes an improvement to an existing method for estimating the mRNA stability. Specifically, in lack of data collected after transcription blockage or data produced by high-throughput pulse-chase methods, existing methods for estimating mRNA stability are using the difference in log-fold change of exonic and intronic RNA. The improvement in the current method consists of removing a potential bias that is related to the maximum capacity of the pre-mRNA processing machinery and the change in transcription rate. This is a great theoretical exercise and application of this method to human brain data found that RBFox proteins and four microRNAs are intimately involved in Alzheimer's disease.

I think this is great, but I also have some concerns that span both practical aspects of the analysis and how useful this work will be in practice.

Major comments:

1. Let's start with the title. The word "deconvolution" can be confusing when it refers to RNA-seq data, as it is being used extensively to signify the partition of RNA-seq data collected from a heterogeneous cell population to the individual cell types where the sample comes. I suggest replacing this word with an alternative.
2. The author considers $V_{p,max}$ and K_p to be constant. This can and should be tested using RNA-Seq datasets that were obtained from the same samples at different times. Otherwise, its effect on the model should be assessed.
3. The author claims that there is significant correlation between $\Delta_{exon} - \Delta_{intron}$ vs Δ_{intron} . This is true because the p-value is very small (see Suppl Fig. 1), but this correlation only explains 14% of the variance. So, it is not clear if the effect is that strong.
4. None of the predicted miRNA sites was validated.
5. The author used HITS-CLIP data from mouse brain as validation. Since all other methods and

datasets used are from human, I am wondering why they did not use human HITS-CLIP data.

6. The overall model performance is not that great. The author states that the 6 factor they found explain "considerable fraction of the mRNA stability profile of the brain". However, the Pearson correlation is 0.31, and although statistically significant, it only explains less than 10% of the observed variance. This is also expected since we know that individual miRNAs have usually a small effect in the expression levels of their target mRNAs unless they have multiple targets on the 3'UTR, or their action is combined with other miRNAs. In either case, no data is shown to justify a significant effect.

7. The author should compare the performance of this method to the existing methods without the bias correction.

Overall assessment.

I like the theoretical aspect of this work and I believe it helps us understand the biases of these models. However, the data presented are not convincing that the bias correction contributes a significant practical improvement. Furthermore, with the availability of accurate biochemical methods to monitor mRNA stability, make this method less important in practice.

Reviewer #3 (Remarks to the Author):

This is an interesting manuscript that identifies a mathematical error in the way mRNA is calculated from large databases using informatics approaches. This error introduces a systematic bias into the analyses, which have purportedly been corrected with the new mathematical algorithm presented in this manuscript. This algorithm is then applied to the case of Alzheimer's disease, where the author identifies RBFOX1 as a RNA binding protein that potentially regulates mRNA stability in the AD brain.

The results are quite interesting, and the pictures are appealing, however I have a number of significant methodological concerns with the manuscript. The most important issue is that the manuscript presents no experimental validation of the purported results. Without such validation, the paper represents an incremental advance to the field. Major concerns are described below:

1. As mentioned, the manuscript lacks any experimental validation for the data. The authors should experimentally compare mRNA stability for a variety of transcripts under the conditions addressed in the manuscript.

2. For instance, the RBFOX stability targets are 2.8 fold more likely to be bound by RBFOX, but is there experimental evidence that RBFOX modulates mRNA stability in the brain

3. Although statistical analysis is at the basis of the studies, the manuscript does not describe these approaches. Thus this information MUST be explicitly provided.

4. The manuscript does not address decay of premRNA through the nuclear exosome pathway. This can lead to differential decay that occurs due to biological variance in premRNA decay independently of any mathematical bias in the methods for calculating mRNA decay.

5. The significance of the conclusions related to regulation by RNA binding proteins is suspect because there the predicted consensus binding preferences of RBPs differ extensively from their actual binding preferences. Similar considerations apply to miRs. So, this gets back to the original comment, which that the lack of experimental validation of the conclusions significantly weakens this manuscript.

6. Is there any experimental evidence from the literature supporting the hypothesis that miR 124, 29 and 128 are associated with regulating mRNA stability?

7. For the miR124 studies the authors jump to HeLa cells, which raises suspicion that the studies didn't work in the brain. Why don't they look at brain, like they did for RBFOX; HeLa cell biology shares little similarity with AD brain.

8. The axes in Fig. 1C need to have labels.

Please find below our point-by-point responses to referee's comments. Our responses are in blue. In the manuscript file, we have also highlighted the major sections of the text that have been modified in this revision compared to the previous version.

Reviewers' comments:

Reviewer #1 (Remarks to the Author):

In this manuscript, Najafabadi proposed a novel method to infer changes in mRNA decay rate from RNA-Seq data by separate analysis of exonic vs. intronic reads. This method is an extension of a previous method published by Gaidatzis et al., based on a kinetic model relating transcription, pre-RNA splicing and RNA delay. Using this method, the author inferred mRNA stability in human tissues including the brain, which was then correlated with binding sites of RBPs and miRNAs. Interestingly, two RBPs (RBFOX1 and ZFP36) and 4 miRNAs were identified as major regulators. In addition, the author also found that RBFOX1 is downregulated in AD brains, which is correlated with destabilization of RBFOX targets. As a method paper, the idea is interesting and the results appear to be promising, but it is currently limited by analysis of single dataset, so the advantage and robustness of the method are not fully demonstrated. More caution is required to interpret the observations from AD brains.

We appreciate that the reviewer has found our work of interest. Below, we have outlined the new analyses and experiments that we have done to address the concerns the reviewer has raised.

Major concerns:

Since this work aims to improve a previous method, a critical point is to demonstrate that the new method with bias correction outperforms the original method. In this regard, the authors made an interesting observation on the negative correlation between ΔE_{delta} vs ΔE_{delta} . The kinetic model appears to make sense, despite some assumptions and simplification not clearly justified. The inference of the gene-specific bias is based on the assumption that there is no change in degradation, which does not seem to be reasonable (Fig. 1c,d). The author only mentioned this in the Fig. 1c legend. This should really be clearly described in Method and justified.

We did not in fact assume that there is no change in degradation. What Figure 1c shows is that even in the absence of any change in degradation, ΔE_{delta} would still be non-zero depending on transcription rate, because of the 'bias term' that we have identified. We have now modified the text to avoid this confusion:

Page 3: "*even in the absence of any change in mRNA stability, ΔE_{delta} would still be positive for genes that were transcriptionally down-regulated, and negative for genes that were transcriptionally up-regulated*".

However, in order to be able to estimate the gene-specific bias terms, we use regression of $\Delta e^{-\Delta i} \sim f_{bias}(\Delta i)$, where f_{bias} is a linear function of Δi . This regression implicitly assumes that the change in mRNA decay constant, i.e. $\log(k_d/k_d')$, is independent of the bias term, which in turn depends on Δi . This assumption enables us to estimate the bias term from regression. In the absence of a clear picture of how transcription and mRNA decay are coupled and the extent to which they are correlated across tissues, we think this is a reasonable simplification, without which this model cannot be implemented. We have now mentioned this implicit assumption in Figure 2a legend, and in the Methods section:

Page 11: "... we estimated the function f_{bias} for each gene by least-square linear regression of $\Delta_{exon}-\Delta_{intron}$ vs. Δ_{intron} across all tissues/samples in each dataset, assuming that the change in mRNA decay constant across samples is independent of the bias term, and therefore the partial regression coefficient in the model $\Delta e^{-\Delta i} = \Delta \log(k_d) + f_{bias}(\Delta i)$ is equal to the regression coefficient in the model $\Delta e^{-\Delta i} \sim f_{bias}(\Delta i)$ ".

The only direct comparison with the previous method was presented in Fig. S3 and the author claimed a 32% improvement in Pearson correlation (from ~0.25 to ~0.34) between predicted and experimentally measured differential stability. This reviewer noticed that the correlation reported by Gaidatzis et al. is 0.29, and it is unclear why is the difference, which will favor the authors' claim. Also, it will be helpful to show the correlation of mRNA stability with and without bias correction so that readers can get an intuitive idea how the correction affects the results.

We have now added the Pearson correlation values to the legend of Supplementary Figure 5 (previously 3). As the reviewer has noted, the value shown in this figure (0.26) is lower than the value reported by Gaidatzis et al. (0.29). We note that, as Supplementary Figure 3a and b show, our framework initially filters the genes with low read counts, and retains only genes that pass a minimum read count cutoff. Therefore, genes shown in Supplementary Figure 5 are only a subset of those reported by Gaidatzis et al. These genes mostly correspond to more abundant transcripts (or longer transcripts), which occupy more of the processing machinery. As equations in Figure 1b (and the graph in Figure 1c) show, such genes have larger bias terms (which can be empirically seen in the RNA-seq data, as shown in Figure 1e). Therefore, for such subset of genes, the performance of $\Delta_{exon}-\Delta_{intron}$ is relatively lower than the entire set of genes. We have now mentioned this in the legend of Supplementary Figure 5:

"Note that REMBRANDTS filters against genes with low read counts (Supplementary Figure 3a), and therefore the set of genes used in this figure are a subset of those used in ref 3. Large read counts are overall associated with larger bias terms (Figure 1c,e), resulting in a smaller correlation between $\Delta_{exon}-\Delta_{intron}$ and mRNA half-life compared to values reported in ref 3."

And also on Page 3 of the manuscript:

“In fact, higher expression was associated with larger magnitudes of bias (Figure 1e), consistent with our kinetic model in which saturation of the pre-mRNA processing machinery results in a steeper bias (Figure 1c).”

Given the critical importance to demonstrate the advantage of the revised method, this reviewer feels it is essential to show the improvement is robust. While I am not aware whether there are other datasets with both RNA-Seq and mRNA stability measurements for the same set of samples, additional direct comparison would be very helpful.

We have now provided additional analyses and experiments to address this concern:

- First, we identified another dataset with both RNA-seq and mRNA stability measurements (Goodarzi et al., 2014), corresponding to the human breast cancer cell line MDA and a highly metastatic sub-line called MDA-LM2. We applied our method to the RNA-seq data from these cell lines, and evaluated the results against the experimentally measured mRNA stabilities. As the new Figure 2b shows, our method clearly provides more accurate estimates for genes that were reported by Goodarzi et al. to be differentially regulated at the stability level between these two cell lines. This improvement is particularly obvious for genes that have the largest bias term (toward the left on each graph in Figure 2b), which is consistent with our model.
- Secondly, we experimentally measured the stability of six additional genes in these two cell lines; these genes were specifically selected from those that had conflicting predictions based on our method and the previous method. In four out of six cases, our qRT-PCR measurements were consistent with the predictions of our method, but not with uncorrected Δ exon– Δ intron. The other two genes did not show statistically significant differential stability. These results are presented in Figure 2c.

It will also be important to analyze an independent RNA-Seq dataset, e.g., derived from mouse tissues, to see whether the bias shown in Fig. 1d, as well as estimates of gene-specific stability in the same tissues, is reproducible. The use of mouse tissues is also handy to avoid complication of RNA quality issues inherent in post-mortem human samples.

We have now provided the distribution of the gene-specific biases for five additional datasets, two representing human RNA-seq data and three representing mouse RNA-seq data. As shown in Supplementary Figure 2a and b, all these datasets clearly show the same bias pattern, with Δ exon– Δ intron being overall negatively correlated with Δ intron for a large majority of genes.

We have also used mouse tissue RNA-seq data to estimate tissue-specific stability profiles, and compared them to stability profiles obtained from human RNA-seq datasets. As shown in Supplementary Figure 7, similar tissues from mouse and human cluster together based on their stability profiles.

I have more reservations on the interpretation of the results from AD brains. I think the result presented in this paper is at best correlative, so claims such as “RBFOX-mediated stabilization of transcripts that encode synaptic transmission proteins is a critical factor for proper functioning of the brain, and defects in this regulatory program can contribute to the onset or progression of AD.” should be avoided. An alternative interpretation of the results is the neuronal loss in advanced AD patients, since RBFOX1 and many transcripts identified by the author as RBFOX1 targets are neuron-specific.

We have made several changes to the manuscript to address this issue.

- We have now provided additional analyses to show that, while neuronal loss is in fact a confounding factor as the reviewer has suggested, it does not fully explain our observations. As shown in the new Supplementary Figure 13, RBFOX stability targets are significantly more down-regulated in AD compared to other neuron-specific genes, suggesting that neuronal loss alone cannot explain this observed down-regulation. Furthermore, RBFOX targets that are not neuron-specific are also down-regulated in AD (neuronal loss alone would not greatly affect the expression measurements of these genes, since they are also expressed in other cells). In addition, RBFOX1 itself is more strongly down-regulated in AD compared to a panel of genes that are almost exclusively expressed in neuron cells (now shown in Supplementary Figure 15b).
- We also note that a rare heterozygous deletion spanning the RBFOX1 locus has been previously connected to early-onset familial AD (Hooli et al., Mol Psychiatry 2014 19:676-81), as now mentioned on page 7 of the manuscript. This is in line with our observation that RBFOX1 down-regulation may be associated with AD.
- However, we completely agree with the reviewer that our results do not provide direct evidence for a causal link between RBFOX1 and AD – providing such direct evidence is in general very challenging in association studies. We have now made sure that we avoid such an interpretation in the manuscript. Below are a few examples of the changes that we've made in the manuscript to ensure that our interpretations are directly supported by our observations:
 - o Page 6: “*These results suggest that RBFOX-bound transcripts that encode synaptic transmission proteins are destabilized in AD.*”
 - o Page 6: “*This suggests that RBFOX1 down-regulation may contribute to the deregulation of the RBFOX stability program in AD.*”
 - o Page 7: “*... suggesting a role of RBFOX1 in AD-associated changes in neural cells.*”

The age/gender difference of RBFOX1 expression in normal brains is not convincing (Fig. 4c,f) given the magnitude of the difference and the potential complications such as RNA quality.

We now mention on page 7 of the manuscript that the association between RBFOX1 expression and age/gender “*can only partially explain the large magnitude of differences that are seen between AD and healthy individuals (Figure 5c,f), consistent with the notion that AD is a complex and multi-factor disease.*” We would also like to point out that age and gender have only a partial association with the likelihood of the onset of AD. Therefore, even in the hypothetical situation that RBFOX1 expression was 100% associated with AD, we would still expect a marginal association between RBFOX1 expression and age/gender.

The siRNA KD data used in Fig. 4d is not reliable, since RBFOX1 expression is very low even in control samples. This is likely due to poor differentiation of the cells. The poor quality of this dataset is also clear from Fig. 2 of the original Fogel paper, since the shGFP samples clusters together with shRBFOX1, instead of WT. This dataset should not be used to avoid misleading readers and contamination of the literature.

We repeated this analysis with Rbfox1/3 knock-down data from mouse neurons (Lee et al., 2016), since we could not find any other shRBFOX1 dataset from human neuron cells. We indeed observed a very strong enrichment of AD-destabilized RBFOX1 targets among genes that are down-regulated after knock-down of Rbfox1/3 in mouse neurons. Furthermore, neuron cells with rescued Rbfox1 expression show the reverse pattern. We have now included these analyses in Supplementary Figure 16 of the revised manuscript.

Minor issues:

There appears to be some inconsistency in reference numbers. For example, Ref. 9 is not the paper with 20 human tissues.

Ref. 10 (previously ref 9) indeed describes the RNA-seq analysis of 20 human tissues, in addition to analysis of Drosophila splicing events. The human RNA-seq data are explained in Extended Data Table 5 of ref. 10.

Fig. 4f, green dots in the figure, but red dots in the legend.

We have now fixed this mistake.

Online Methods, additional explanation of the kinetic model will be helpful (e.g., why is the addition of $V_t=V_p=V_d$? What choose Michaelis-Menten kinetics to model splicing?). The author might do a better job to put the model into similar work in the literature?

We have now added additional descriptions to the section “Modeling the pre-mRNA and mRNA abundance” in order to clarify the choice of kinetic model and underlying assumptions. We have also included a new Supplementary Figure 1 that fully explains our model and calculations.

Fig. S1 legend, “...results in an apparent anti-correlation between transcriptional and post-transcriptional regulation of pathways”. This is certainly not true for all panels.

We apologize for the incomplete figure legend. The figure legend has now been corrected to clarify that the anti-correlation corresponds only to panel c, while panel d shows an instance in which the correlation between transcriptional and post-transcriptional regulation of pathways increases as a result of bias correction.

Related to Fig. 4c,e and Fig. S9, it will be helpful to show the level of RBFOX2/3 from microarray data as well, and show the abundance of RBFOX1/2/3 estimated from exonic reads in RNA-Seq data.

This information has now been added to Supplementary Figure 15.

Reviewer #2 (Remarks to the Author):

This paper describes an improvement to an existing method for estimating the mRNA stability. Specifically, in lack of data collected after transcription blockage or data produced by high-throughput pulse-chase methods, existing methods for estimating mRNA stability are using the difference in log-fold change of exonic and intronic RNA. The improvement in the current method consists of removing a potential bias that is related to the maximum capacity of the pre-mRNA processing machinery and the change in transcription rate. This is a great theoretical exercise and application of this method to human brain data found that RBFOX proteins and four microRNAs are intimately involved in Alzheimer's disease.

We appreciate the positive assessment of the reviewer regarding the significance of this work.

I think this is great, but I also have some concerns that span both practical aspects of the analysis and how useful this work will be in practice.

Major comments:

1. Let's start with the title. The word "deconvolution" can be confusing when it refers to RNA-seq data, as it is being used extensively to signify the partition of RNA-seq data collected from a heterogeneous cell population to the individual cell types where the sample comes. I suggest replacing this word with an alternative.

We agree with the reviewer. We have now changed the title to avoid any confusion with deconvolution of mixed cell RNA-seq data.

2. The author considers $V_{p,max}$ and K_p to be constant. This can and should be tested using RNA-Seq datasets that were obtained from the same samples at different times. Otherwise, its effect on the model should be assessed.

We have now addressed this issue using the following approaches:

- First, we solved the kinetic equations of mRNA metabolism without assuming that $V_{p,max}$ and K_p are constant. This revealed that in this situation, the relationship between $\Delta_{\text{exon}}-\Delta_{\text{intron}}$ and Δ_{intron} would depend on the fold-change in $V_{p,max}$, and is also a function of K_p (Supplementary Figure 4a). This equation suggests that when the change in $V_{p,max}$ is considerably smaller in magnitude than the change in degradation rate, its effect is negligible and can be ignored. Similarly, if the change in K_p is considerably smaller than the change in the concentration of pre-mRNA, our assumption works well.
- Secondly, we have provided simulations showing that even when these assumptions are violated and both $V_{p,max}$ and K_p vary greatly across samples, our model still provides better estimates of mRNA stability compared to $\Delta_{\text{exon}}-\Delta_{\text{intron}}$ (Supplementary Figure 4b).
- Thirdly, we have modified the manuscript, including the Methods section (page 10) and Supplementary Figure 1, to explicitly mention this assumption and the limitations of our method because of it.

3. The author claims that there is significant correlation between $\Delta_{\text{exon}} - \Delta_{\text{intron}}$ vs Δ_{intron} . This is true because the p-value is very small (see Suppl Fig. 1), but this correlation only explains 14% of the variance. So, it is not clear if the effect is that strong.

The correlation between $\Delta_{\text{exon}}-\Delta_{\text{intron}}$ and Δ_{intron} varies among the genes, and also among datasets. For example, in the 20 human tissues that we analyzed, while the median of R^2 is 0.16, for 25% of the genes R^2 is >0.35 , and for 12% of genes R^2 is >0.5 . For other datasets, a much higher proportion of genes could be affected. For example, in the three mouse datasets that we have added to Supplementary Figure 2b, the median R^2 is >0.63 , indicating that for more than half of the genes, at least 63% of the variance in $\Delta_{\text{exon}}-\Delta_{\text{intron}}$ can be explained by Δ_{intron} alone. This indicates that a large subset of genes could be dramatically affected by this bias. This is also evident in our new analyses in Figure 2b, where for genes that have a large bias slope, uncorrected $\Delta_{\text{exon}}-\Delta_{\text{intron}}$ almost completely fails to predict the change in stability.

4. None of the predicted miRNA sites was validated.

We have now added additional analyses to show that our predicted miRNA targets are enriched for experimentally validated miRNA-mRNA interactions reported in the literature (Figure 4e). This analysis shows that in the case of all the four miRNAs we have analyzed, our predicted high-confidence sites are significantly more likely to be among experimentally validated targets compared to other mRNAs that simply have a match to the consensus binding sequence of the miRNA. In addition to the direct comparison with known targets, we have now also provided indirect evidence for functionality of these binding sites, by showing

that they are significantly more conserved than their immediate adjacent sequences (Supplementary Figure 10).

5. The author used HITS-CLIP data from mouse brain as validation. Since all other methods and datasets used are from human, I am wondering why they did not use human HITS-CLIP data.

The main reason to use mouse HITS-CLIP data for RBFOX proteins was that, to our knowledge, a similar dataset from human neuron cells was not available. We note that RBFOX expression is highly specific to neural cells, limiting us to the analysis of only this particular cell type. We would also like to point out that the analysis provided in Figure 5d of the paper is also an alternative validation of our high-confidence RBFOX targets in human neuronal cells, where we show that these targets are down-regulated after siRNA inhibition of RBFOX1.

6. The overall model performance is not that great. The author states that the 6 factor they found explain “considerable fraction of the mRNA stability profile of the brain”. However, the Pearson correlation is 0.31, and although statistically significant, it only explains less than 10% of the observed variance. This is also expected since we know that individual miRNAs have usually a small effect in the expression levels of their target mRNAs unless they have multiple targets on the 3'UTR, or their action is combined with other miRNAs. In either case, no data is shown to justify a significant effect.

We agree with the reviewer that the interpretation of what is “considerable” could be subjective. Therefore, while we believe that it is remarkable to be able to explain 10% of the variance in stability based on only 6 regulatory factors, we have modified the text to avoid this subjective interpretation. However, we would like to point out that even across different datasets, the correlation of measured mRNA stability could be low ($R^2=0.31$ in the case of comparison of two different brain RNA-seq datasets; please see the new Figure 3f). Therefore, at least part of the nominally small R^2 of the predictions of our model and the measured mRNA stability profile could be attributed to low reproducibility of the RNA-seq data itself (perhaps due to differences in platforms, handling of post-mortem tissues, RNA quality, biological variance, and noise). This is now mentioned in the manuscript at the end of page 4:

“Together with rapid decay of non-neural transcripts by miRNAs, this parsimonious regulatory model, which consists of only six regulatory factors (Figure 3e), explains ~10% of the observed variance of the mRNA stability profile of the brain, compared to ~31% of the variance that is reproducible across samples/platforms (Figure 3f).”

We also note that a large fraction of the unexplained variance corresponds to genes that are not differentially stabilized/destabilized in brain (i.e. with stability values around zero). The large number of these genes causes their variance (which is more likely to reflect noise) to dominate the correlation analysis. In contrast to these genes, genes with large stability changes are predicted more accurately by our model: our model can separate genes that are >2-fold up-regulated from those that are >2-fold down-regulated with area under the ROC curve (AUROC) = 0.86, and those that are >4-fold up-regulated from those that are >4-fold down-regulated with AUROC = 0.91 (new Supplementary Figure 9).

7. The author should compare the performance of this method to the existing methods without the bias correction.

In addition to our previous comparison between our method and the uncorrected estimates (based on mouse RNA-seq and mRNA stability measurements, now in Supplementary Figure 5), we have performed additional analyses and experiments, as outlined below:

- We identified another dataset with both RNA-seq and mRNA stability measurements (Goodarzi et al., 2014), corresponding to the human breast cancer cell line MDA and a highly metastatic sub-line called MDA-LM2. We applied our method to the RNA-seq data from these cell lines, and evaluated the results against the experimentally measured mRNA stabilities. As the new Figure 2b shows, our method clearly provides more accurate estimates for genes that were reported by Goodarzi et al. to be differentially regulated at the stability level between these two cell lines. This improvement is particularly obvious for genes that have the largest bias term, which is consistent with our model.
- We also experimentally measured the stability of six additional genes in these two cell lines; these genes were specifically selected from those that had conflicting predictions based on our method and the previous method. In four out of six cases, our qRT-PCR measurements were consistent with the predictions of our method, but not with uncorrected Δ exon- Δ intron. The other two genes did not show statistically significant differential stability. These results are presented in Figure 2c.

Overall assessment.

I like the theoretical aspect of this work and I believe it helps us understand the biases of these models. However, the data presented are not convincing that the bias correction contributes a significant practical improvement. Furthermore, with the availability of accurate biochemical methods to monitor mRNA stability, make this method less important in practice.

We appreciate that the reviewer has found our theoretical work interesting, and we hope that our new analyses and experiments, as described above, now provide convincing support that bias correction is necessary to estimate mRNA stability from RNA-seq data. We would like to point out that while biochemical methods are now available to monitor genome-wide mRNA stability in cell culture, as the reviewer has mentioned, we are not aware of any method that can provide direct measures of mRNA stability in tissue samples, e.g. biopsy samples from patients, or human post-mortem samples. Therefore, inferring stability from

transcription in RNA-seq data provides one of the few methods, if not the only applicable method, to specifically study mRNA stability in such samples.

Reviewer #3 (Remarks to the Author):

This is an interesting manuscript that identifies a mathematical error in the way mRNA is calculated from large databases using informatics approaches. This error introduces a systematic bias into the analyses, which have purportedly been corrected with the new mathematical algorithm presented in this manuscript. This algorithm is then applied to the case of Alzheimer's disease, where the author identifies RBFOX1 as a RNA binding protein that potentially regulates mRNA stability in the AD brain.

The results are quite interesting, and the pictures are appealing, however I have a number of significant methodological concerns with the manuscript. The most important issue is that the manuscript presents no experimental validation of the purported results. Without such validation, the paper represents an incremental advance to the field.

We are glad the reviewer has found this manuscript interesting. We have now performed additional experiments and analyses to address the issues that the reviewer has pointed out, as outlined below.

Major concerns are described below:

1. As mentioned, the manuscript lacks any experimental validation for the data. The authors should experimentally compare mRNA stability for a variety of transcripts under the conditions addressed in the manuscript.

We have now performed this experiment, in addition to complementary analyses to show that our method provides more reliable estimates of mRNA stability compared to existing methods. Specifically:

- We had previously presented an analysis showing that in a mouse dataset with both RNA-seq and mRNA stability measurements, our method provided reliable stability estimates. We have now identified another dataset with both RNA-seq and mRNA stability measurements (Goodarzi et al., 2014), corresponding to the human breast cancer cell line MDA and a highly metastatic sub-line called MDA-LM2. We applied our method to the RNA-seq data from these cell lines, and evaluated the results against the experimentally measured mRNA stabilities. As the new Figure 2b shows, our method clearly provides more accurate estimates for genes that were reported by Goodarzi et al. to be differentially regulated at the stability level between these two cell lines.
- We also experimentally measured the stability of six additional genes in these two cell lines; these genes were specifically selected from those that had conflicting predictions based on our method and the previous method. In four out of six cases, our qRT-PCR measurements were consistent with the predictions of our method, but not with uncorrected Δ exon- Δ intron. The other two genes did not show statistically significant differential stability. These results are presented in Figure 2c.

2. For instance, the RBFOX stability targets are 2.8 fold more likely to be bound by RBFOX, but is there experimental evidence that RBFOX modulates mRNA stability in the brain

A few recent publications have suggested a role of RBFOX1 in regulating mRNA stability. For example, Ray et al. (2013) performed a series of analyses, in addition to reporter assays, showing that RBFOX1 binding to the 3' UTR increases the mRNA abundance. Specifically, they showed that “the mRNA abundance of a reporter construct harbouring a single RBFOX1 site in the 3' UTR increased, relative to an identical reporter containing a mutant RBFOX1 site, upon induction of RBFOX1 expression”. These publications have been cited in the manuscript, e.g. on page 4: “... *previous studies suggest that RBFOX proteins stabilize their target mRNAs ...*”.

3. Although statistical analysis is at the basis of the studies, the manuscript does not describe these approaches. Thus this information MUST be explicitly provided.

We have now ensured that all the statistical analyses are properly explained in the manuscript. These include description of the statistical tests used to obtain all the P -values, as well as complete representation of the equations that underlie our kinetic model (please see the new Supplementary Figure 1). In addition to the detailed descriptions of the methods in this manuscript, we have deposited the scripts and the data that are required for reproduction of our main results in the GitHub repository (<https://github.com/csglab/REMBRANDTS>).

4. The manuscript does not address decay of premRNA through the nuclear exosome pathway. This can lead to differential decay that occurs due to biological variance in premRNA decay independently of any mathematical bias in the methods for calculating mRNA decay.

We have now added new calculations, which are fully presented in the new Supplementary Figure 1, showing that addition of a step for decay of pre-mRNA does not change the relationship between $\Delta_{\text{exon}} - \Delta_{\text{intron}}$ and Δ_{intron} . This counterintuitive result can be understood by imagining an “apparent rate of production” for pre-mRNA, which equals the rate of transcription minus the rate of decay of the pre-mRNA. It is then easy to see that the calculations in Supplementary Figure 1a do not change after introducing pre-mRNA decay, and only the transcription rate (V_i) is replaced by this apparent rate of production.

5. The significance of the conclusions related to regulation by RNA binding proteins is suspect because there the predicted consensus binding preferences of RBPs differ extensively from their actual binding preferences. Similar considerations apply to miRs. So,

this gets back to the original comment, which that the lack of experimental validation of the conclusions significantly weakens this manuscript.

We share this concern with the reviewer that the consensus binding preferences of RBPs are poor predictors of their actual binding sites, and often result in a large number of false positives. As we now describe in the first paragraph of the Discussion section, the main motivation behind creating a high-confidence network based on combination of consensus sequences and mRNA stability was in fact to reduce these false positives and identify functional targets of RBPs and miRNAs. Indeed, as we show in Figure 4, for all of the six factors that we examined, our high-confidence network is enriched for previously known targets compared to transcripts that only match the consensus binding sequence. Interestingly, our new analyses show that the high-confidence binding sites of RBPs have structural properties consistent with what we would expect for binding of protein (Supplementary Figure 17) – One of the main reasons that the consensus binding sequences cannot reliably identify the binding sites of RBPs is the effect of RNA structure on the accessibility of the binding sequence; we now show that although we do not use any information from RNA structure, our high-confidence binding sites are indeed more accessible.

6. Is there any experimental evidence from the literature supporting the hypothesis that miR 124, 29 and 128 are associated with regulating mRNA stability?

We have now added citations to several previous publications showing that the binding of these miRNAs reduces mRNA abundance (page 4: “*All four miRNAs have been previously shown to be able to decrease the abundance of their target transcripts* ^[26-30], suggesting that they can indeed destabilize their targets.”).

7. For the miR124 studies the authors jump to HeLA cells, which raises suspicion that the studies didn't work in the brain. Why don't they look at brain, like they did for RBFOX; HeLa cell biology shares little similarity with AD brain.

Since miR-124 is already expressed at high levels in the brain and its targets are already inhibited, over-expression of miR-124 in neural cells would not result in a tangible effect, which is why we studied available miR-124 over-expression data in HeLa cells (HeLa cells normally do not express miR-124), following several other studies that have similarly studied miR-124 function in non-neural cells [a few examples are: Nam et al., Mol Cell 2014 53(6):1031-43; Hafner et al., Cell 2010 141(1):129-41; Hendrickson et al., PLoS Biol 2009 7(11):e1000238; Eichhorn et al., Mol Cell 2014 56(1):104-15] . While we agree with the reviewer that it would have been preferable to directly examine the effect of miR-124 manipulation in neurons, we were not able to identify any available RNA-seq/microarray

dataset in which miR-124 expression is inhibited in human neuronal cells (possibly due to the many technical challenges that need to be overcome for such an experiment). However, we have now added an additional analysis showing that the collection of the known miR-124 targets that are previously described in the literature are strongly enriched among our high-confidence targets of miR-124. The same is true for the other three miRNAs as well, as we now demonstrate in the new Figure 4e.

8. The axes in Fig. 1C need to have labels.

This is probably the fault of PDF conversion / PDF viewer software, since the axes on Figure 1c do have labels; we apologize for this inconvenience. Please find below an image of this panel:

Reviewers' comments:

Reviewer #1 (Remarks to the Author):

The authors have been very responsible to address my previous questions. I believe the revised manuscript is stronger than the initial version, and I do not have further comments.

Reviewer #2 (Remarks to the Author):

This is a revision of a previous submission, which described an extension of an existing method for estimating the mRNA stability. I am happy to see that the authors tried to address all of the previous comments. I really commend the authors for the additional simulations and experimental validations they performed. However some concerns are still remaining.

Specifically:

1. The authors worked out the math and performed simulations to assess whether $V_{p,max}$ and K_p being constant or not affects their model. This is great, because it showed that when the change in $V_{p,max}$ is considerably smaller in magnitude than the change in degradation rate, its effect is negligible and can be ignored. But the simulations showed that their model still represents somewhat of an improvement. It is too bad that the authors did not take the final step to assess the extent that $V_{p,max}$ and K_p are constant in real datasets, which I thought would have been much easier.

2. In my original comment that the correlation between $\Delta_{exon} - \Delta_{intron}$ vs Δ_{intron} only explains 14% of the variance they responded that the correlation varies between the genes and for 12% of the genes R^2 is >0.5 . But this is true for any correlation comparison. One can always find a subset of the pairs where correlation is much stronger than the full dataset. This however does not make the method any better. Their Suppl Fig 2 also attest to that.

3. Following mine and other reviewers comment to compare their new method to existing ones the authors used a new dataset and performed qPCR validation (Figure 2). First, it is not clear why they did not compare their method to existing ones on the Alzheimer dataset, since this is central to their paper. But I also found Fig 2 confusing. For example in Fig 2b the authors show that the bias correction separates better the stabilized from the destabilized mRNAs than the method that does not include bias correction. But this separation is better in all bias thresholds, positive or negative, even at the very beginning. This is counterintuitive and does not agree with the fact that only a small percent of genes exhibit this bias. It is counterintuitive because when bias=0 one would expect that the two methods will perform the same. But we see that there is a big gap that separates the red and the blue points in the right panel (new method) that is not present in the left panel (existing method).

4. In the same figure they also report new experimental validation of 6 genes, which is commendable. They do not explain how they selected these genes, but I assume these were their top 6 predictions. In other words these are most likely the genes in which their predicted bias term was the highest. This is the range in which, as they said, their method can explain most of the variance. Two of those 6 predictions (33%) fail to show any significant change. Besides, neither in this section nor in the Methods the authors explain how exactly they calculated the p-value.: what values did they use for the two-tailed Mann-Whitney test? $\Delta\Delta Ct$ or something else?

Reviewer #3 (Remarks to the Author):

This is a resubmitted manuscript that presents an advance in identifying destabilization of RNA transcripts, with a particular focus on Alzheimer's disease. The authors responded to many of the comments robustly, by adding new datasets to the analyses and moderating the text to reflect limitations in the methodology. In the end the authors state that the data appear to be able to account for about 10% of the decrease of RNA stability in AD. This aspect of the work seems to apply well to multiple different data sets. The authors show that it also extends to AD brain. However, interpretation for AD remains a challenge. RBFOX is a neuron specific protein; in AD cortex about 30% of neurons are lost, and other neurons are in the process of degenerating. Thus it is not clear how much of the changes are due to neuronal loss vs. downregulation of RBFOX. The authors state that "RBFOX targets that are not neuron-specific are also downregulated in AD (neuronal loss alone would not greatly affect the expression measurements of these genes, since they are also expressed in other cells)." It is unclear why RBFOX regulation would be the same in neuronal and nonneuronal cells.

The statistics and mathematical modeling appear to be acceptable.

However, when considering the paper in total, I suspect that the findings are real, and that this paper provides a modest advance to the field.

Please find our responses to the referee comments below. Our responses are in blue. An itemized list of changes to the manuscript can be found at the end of this document.

Reviewers' comments:

Reviewer #1 (Remarks to the Author):

The authors have been very responsible to address my previous questions. I believe the revised manuscript is stronger than the initial version, and I do not have further comments.

We are glad that Reviewer #1 has found our responses satisfactory.

Reviewer #2 (Remarks to the Author):

This is a revision of a previous submission, which described an extension of an existing method for estimating the mRNA stability. I am happy to see that the authors tried to address all of the previous comments. I really commend the authors for the additional simulations and experimental validations they performed. However some concerns are still remaining.

We thank Reviewer #2 for the positive assessment of our revised manuscript. We hope that our additional explanations (below) and the modifications we have made in the manuscript will address the remaining concerns:

Specifically:

1. The authors worked out the math and performed simulations to assess whether $V_{p,max}$ and K_p being constant or not affects their model. This is great, because it showed that when the change in $V_{p,max}$ is considerably smaller in magnitude than the change in degradation rate, its effect is negligible and can be ignored. But the simulations showed that their model still represents somewhat of an improvement. It is too bad that the authors did not take the final step to assess the extent that $V_{p,max}$ and K_p are constant in real datasets, which I thought would had been much easier.

We believe measuring the variables $V_{p,max}$ and K_p across tissues is not a trivial task, and would require experimental methods that, to our knowledge, currently do not exist. Obtaining such measurements generally requires manipulation of transcription rate of specific genes followed by measuring the mRNA processing rate, which would be extremely cumbersome even in cell culture, and prohibitively complicated in live tissues. We are also not aware of analytical methods to directly measure $V_{p,max}$ and K_p of each gene across tissues based on existing data.

However, as the reviewer has noted, we have tried to address this limitation by showing that even when $V_{p,max}$ and K_p vary widely across tissues, our method is still more accurate than the previous method (Supplementary Figure 4b). This notion is also supported by comparison of our new method and the previous method based on real data (Figure 2 and Supplementary Figure 5).

Therefore, even if it was possible to measure $V_{p,max}$ and K_p across tissues, we would not expect to see any change in the main conclusions of this paper, in that bias-correction considerably improves estimates of mRNA decay rate.

2. In my original comment that the correlation between $\Delta_{exon} - \Delta_{intron}$ vs Δ_{intron} only explains 14% of the variance they responded that the correlation varies between the genes and for 12% of the genes R^2 is >0.5 . But this is true for any correlation comparison. One can always find a subset of the pairs where correlation is much stronger than the full dataset. This however does not make the method any better. Their Suppl Fig 2 also attest to that.

While it is true that in any correlation comparison there is variation because of pure chance, the extent of this variation is expected to be limited. For example, when we are measuring the correlation of two variables with 20 data points (e.g. the correlation of $\Delta_{exon} - \Delta_{intron}$ and Δ_{intron} across 20 tissues), we would expect to see only 0.02% of genes to have $R^2 > 0.5$ due to chance (compared to 12% that we observed for the human RNA-seq data from 20 tissues). We also note that in other datasets that we analyzed, we observed much larger biases (Supplementary Figure 2a,b).

We have now modified Supplementary Figure 2a-b to emphasize that a large fraction of genes in each of the RNA-seq datasets that we analyzed showed a bias that would not be expected by chance. In each histogram, the blue fraction corresponds to the genes with significant negative Pearson correlations ($FDR < 0.05$) – we would expect that only 5% of the blue fraction would show the observed bias by chance.

Furthermore, our analysis of experimental stability data indicates that for a large fraction of genes, bias correction provides substantial improvement for measuring mRNA stability (Figure 2b). Therefore, we maintain our initial conclusion that bias correction is essential for inferring mRNA stability from RNA-seq.

3. Following mine and other reviewers comment to compare their new method to existing ones the authors used a new dataset and performed qPCR validation (Figure 2). First, it is not clear why they did not compare their method to existing ones on the Alzheimer dataset, since this is central to their paper.

We were limited by the availability of data and methods to measure mRNA stability: in order to show that bias-correction provides more accurate estimates of mRNA stability, we needed to compare the estimates of our method with experimentally measured mRNA stability data, which are not available for Alzheimer's datasets. To our knowledge, current experimental methods do not allow direct measurement of mRNA stability in post-mortem tissue samples; therefore, we could only validate the performance of our method based on stability measurements in cell lines.

But I also found Fig 2 confusing. For example in Fig 2b the authors show that the bias correction separates better the stabilized from the destabilized mRNAs than the method that does not include bias correction. But this separation is better in all bias thresholds, positive or negative, even at the very beginning. This is counterintuitive and does not agree with the fact that only a small percent of genes exhibit this bias. It is counterintuitive because when bias=0 one would expect that the two methods will perform the same. But we see that there is a big gap that separates the red and the blue points in the right panel (new method) that is not present in the left panel

(existing method).

This observation would be counterintuitive if, as the reviewer has mentioned, only a small percent of genes exhibited bias. However, as we noted in our response above, this is not necessarily the case, and a large fraction of genes do show a substantial bias in many datasets, including the one presented in Figure 2b. Please note that in Figure 2b, as mentioned in the legend, "*each point represents the set of genes that have a bias slope smaller than the corresponding cutoff on the x-axis*". Therefore, the point at $x=0$ represents the set of genes with bias <0 , including genes with highly negative bias.

We have now added an additional panel to Supplementary Figure 5 (panel a), showing that for the set of genes we analyzed in Figure 2b (which were selected based on previous BRIC-seq results), a large fraction shows considerable amount of bias. Also, the new panel b in Supplementary Figure 5 provides an alternative presentation of the same data as in Figure 2b. This new panel shows that, as expected, the two methods perform the same for genes with biases near zero, while our method substantially improves the stability measurements for highly biased genes.

4. In the same figure they also report new experimental validation of 6 genes, which is commendable. They do not explain how they selected these genes, but I assume these were their

top 6 predictions. In other words these are most likely the genes in which their predicted bias term was the highest.

The reviewer is correct in that these six genes had the highest predicted bias. The details of how these genes were selected were described in the methods section:

Page 11: *“We used RNA-seq data from ref 15 to measure uncorrected and bias-corrected Δ exon– Δ intron, and then identified genes that were inferred to be up-regulated or down-regulated in MDA-parental or MDA-LM2 cells based on each measure (two-tailed Student’s t-test $P < 0.05$). We then excluded all genes with previously reported differential stability scores > 0.5 or < -0.5 between MDA-parental and MDA-LM2 cells [ref 17], and among the remaining genes, identified those that had conflicting predictions from uncorrected and bias-corrected Δ exon– Δ intron. We sorted these genes based on their bias slopes, and selected the three most highly biased genes that were predicted by uncorrected Δ exon– Δ intron to be stabilized in MDA-LM2, as well as the three most highly biased genes that were predicted based on uncorrected Δ exon– Δ intron to be destabilized in MDA-LM2 relative to MDA-parental”*.

We have now also modified the main text to mention that these six genes had the highest bias:

Page 3: *“To further evaluate our method, we selected six additional genes that did not have statistically significant differences in BRIC-seq measurements between MDA-parental and MDA-LM2 cells, and had the largest bias based on analysis of RNA-seq data (see Methods)”*.

This is the range in which, as they said, their method can explain most of the variance. Two of those 6 predictions (33%) fail to show any significant change.

The two genes that did not show statistically significant differences were predicted by both our method and the previous method to have differential stability. In other words, for these two genes, our method and the previous method were both wrong, and for the other four genes, only our method correctly predicted the direction of change in stability. Therefore, our method was able to infer the stability correctly in 4 out of 6 cases, and the previous method correctly inferred the stability in 0 out of 6 cases.

Besides, neither in this section nor in the Methods the authors explain how exactly they calculated the p-value.: what values did they use for the two-tailed Mann-Whitney test? $\Delta\Delta$ Ct or something else?

We apologize for omitting this detail in the previous submission; we have now added it to the legends of Figure 2c:

*“Significant differences between the two cell types are marked with asterisks (*Mann-Whitney U test on relative stabilities, $P < 0.05$)”.*

Relative stability is defined in the legends of Figure 2c as well as in the Methods section.

We should point out that since Mann-Whitney U test is non-parametric, the results would be identical if another measure was used instead of relative stability, such as $\Delta\Delta Ct$.

Reviewer #3 (Remarks to the Author):

This is a resubmitted manuscript that presents an advance in identifying destabilization of RNA transcripts, with a particular focus on Alzheimer's disease. The authors responded to many of the comments robustly, by adding new datasets to the analyses and moderating the text to reflect limitations in the methodology.

We are happy that Reviewer #3 has found our responses to the previous concerns robust.

In the end the authors state that the data appear to be able to account for about 10% of the decrease of RNA stability in AD. This aspect of the work seems to apply well to multiple different data sets. The authors show that it also extends to AD brain. However, interpretation for AD remains a challenge. RBFOX is a neuron specific protein; in AD cortex about 30% of neurons are lost, and other neurons are in the process of degenerating. Thus it is not clear how much of the changes are due to neuronal loss vs. downregulation of RBFOX.

We have now acknowledged this limitation in the manuscript, explicitly mentioning that despite our efforts to exclude the possibility that neuronal loss affects our conclusions, it is still a confounding factor in our analyses:

Page 6: *“Given that many of RBFOX targets are specifically expressed in neurons, one possibility is that down-regulation of these transcripts reflects neuronal loss, which is commonly seen in advanced AD [ref 36]. However, we observed that RBFOX targets were significantly more down-regulated compared to other neuron-specific genes ...”*

Page 8: "Furthermore, interpretation of AD transcriptomics data remains challenging given the confounding effect of neuronal loss on gene expression measurements, despite our attempts to control for this confounding factor in our analyses (Supplementary Figures 13 and 15)."

The authors state that "RBFOX targets that are not neuron-specific are also downregulated in AD (neuronal loss alone would not greatly affect the expression measurements of these genes, since they are also expressed in other cells)." It is unclear why RBFOX regulation would be the same in neuronal and nonneuronal cells.

We should clarify that we are not claiming that RBFOX regulation is the same in neuronal and non-neuronal cells. As explained in page 6, the genes that are shown in Supplementary Figure 13c are expressed in both neurons and non-neuronal cells. Down-regulation of these genes in neuronal cells would show up as overall down-regulation of these genes in the brain transcriptome. On the other hand, a change in the ratio of neuronal and non-neuronal cells (due to neuronal loss) is not expected to change the apparent abundance of these genes in the brain RNA-seq data. The figure below hopefully provides a more intuitive presentation of this problem:

This is now clarified in the manuscript:

Page 6: *“In addition, RBFOX targets whose expression is not limited to neural cells were also down-regulated in AD (Supplementary Figure 13c) – a change in the ratio of neural cells would have a relatively small impact on the apparent abundance of these transcripts since they are also expressed in other cell types, suggesting that the observed destabilization of RBFOX targets is not an artifact of neuronal loss in AD.”*

This analysis was provided as further supporting evidence that our observations are most likely not explained by neuronal loss alone. However, as we mentioned above, we have now added further discussion to the manuscript to emphasize that despite our efforts, the confounding effect of neuronal loss is still a challenge when interpreting AD gene expression data.

The statistics and mathematical modeling appear to be acceptable.

However, when considering the paper in total, I suspect that the findings are real, and that this paper provides a modest advance to the field.

ITEMIZED LIST OF CHANGES

Changes to address referee comments

- **Page 3:** Clarified that the six genes selected for qRT-PCR validation in Figure 2c had the largest bias slopes (in response to Reviewer #2).
- **Pages 6 and 8:** Added “*Given that many of RBFOX targets are specifically expressed in neurons, one possibility is that down-regulation of these transcripts reflects neuronal loss, which is commonly seen in advanced AD*”, and “*Furthermore, interpretation of AD transcriptomics data is particularly challenging given the confounding effect of neuronal loss on gene expression measurements, despite our attempts to control for this confounding factor in our analyses*”(in response to Reviewer #3, to acknowledge that the interpretation of AD data may be confounded by neuronal loss despite our attempts to control for this confounding factor).
- **Page 6:** Modified “*In addition, RBFOX targets whose expression is not limited to neural cells were also down-regulated in AD (Supplementary Figure 13c) – a change in the ratio of neural cells would have less impact on the apparent abundance of these transcripts since they are also expressed in other cell types, suggesting that the observed destabilization of RBFOX targets is not an artifact of neuronal loss in AD*” (to clarify the analyses in Supplementary Figure 13c, in response to Reviewer #3).
- **Legend of figure 2c:** Added: “*Mann-Whitney U test on relative stabilities, $P < 0.05$* ” (in response to Reviewer #2, comment about the missing information with respect to the Mann-Whitney U test).
- **Supplementary Figure 2a,b:** Highlighted the fraction of genes that have a significant bias (FDR<0.05). This was done in response to Reviewer #2, to show that the bias reported in this manuscript is present in a large number of genes, and is significantly more frequent than what is expected by chance.
- **Supplementary Figure 5:** Added two panels (panels a and b) to show the histogram of bias in RNA-seq data from MDA-parental and MDA-LM2 cells, and to show the improvement in stability estimates after bias correction as a function of the bias term (in response to Reviewer #2, comment regarding Figure 2b being counterintuitive). The new panel (a) illustrates that many of the genes presented in Figure 2b show the bias that we reported in this manuscript. Panel (b) is an alternative representation of Figure 2b to show that bias-corrected and uncorrected stability estimates are similar for genes with a small bias term.

Changes to conform to formatting requirements

- **Page 5:** Changed subsection header from “Brain mRNA stability network suggests dysregulation of RBFOX programs in Alzheimer’s disease” to “Dysregulation of RBFOX programs in Alzheimer’s disease” (to conform to formatting checklist item: subtitles must not be more than 60 characters long including spaces).
- **Pages 17:** Relocated “Data and code availability” statement from below the references to below the methods (to conform to formatting checklist item stating that data availability statement must be at the end of methods section).

- **Page 17:** Added a “competing interests” section declaring no competing financial interest (to conform to formatting checklist item requiring a conflict of interest statement in manuscript end notes).
- **Page 11:** Described how RNA binding protein affinities to 3’UTRs were predicted to avoid “as previously described” statement (to conform to formatting checklist item discouraging use of the statement “as previously described” in methods section).
- **Main manuscript, Figures 2, 3, and 5:** Italicized gene/RNA names (to conform to formatting checklist item requiring gene names to be italicized).
- **Legend of figure 3a:** Added definition of error bars.
- **Legend of Supplementary Figure 5d:** Added definition of error bars.
- **Legend of supplementary Figure 17c:** Added definition of error bars.
- **Supplementary information, Page 19:** Change references heading in supplementary information file from “references” to “supplementary references” (to conform to checklist item specifying how supplementary items must be labelled).

REVIEWERS' COMMENTS:

Reviewer #3 (Remarks to the Author):

I have no further criticisms. The reviewers have made changes that improve the manuscript. I am now satisfied that the manuscript conveys a significant advance for the field, and also conveys sufficient nuance to the interpretation.